ecology, evolution

competition, density-dependent, flowering time, deleterious mutations, phenotypic trade-offs, range expansion

**Authors for correspondence:**
Marion Orsucci
e-mail: marion.orsucci@gmail.com
Martin Lascoux
e-mail: martin.lascoux@ebc.uu.se

# Shift in ecological strategy helps marginal populations of shepherd's purse (*Capsella bursa-pastoris*) to overcome a high genetic load

Marion Orsucci[1], Pascal Milesi[1,2], Johanna Hansen[1], Johanna Girodolle[1], Sylvain Glémin[1,3] and Martin Lascoux[1]

[1]Department of Ecology and Genetics, Evolutionary Biology Centre, Uppsala University, Uppsala, Sweden
[2]Science for Life Laboratory, Uppsala, Sweden
[3]ECOBIO UMR 6553 CNRS University of Rennes 1, Rennes, France

MO, 0000-0001-8516-1361; SG, 0000-0001-7260-4573

The outcome of species range expansion depends on the interplay of demographic, environmental and genetic factors. Self-fertilizing species usually show a higher invasive ability than outcrossers but selfing and bottlenecks during colonization also lead to an increased genetic load. The relationship between genomic and phenotypic characteristics of expanding populations has, hitherto, rarely been tested experimentally. We analysed how accessions of the shepherd's purse, *Capsella bursa-pastoris*, from the colonization front or from the core of the natural range performed under increasing density of competitors. First, accessions from the front showed a lower fitness than those from the core. Second, for all accessions, competitor density impacted negatively both vegetative growth and fruit production. However, despite their higher genetic load and lower absolute performances, accessions from the front were less affected by competition than accessions from the core. This seems to be due to phenotypic trade-offs and a shift in phenology that allow accessions from the front to avoid competition.

## 1. Introduction

Species range expansion is a complex process that depends on interplay between demographic, environmental and genetic factors. High dispersal ability is obviously key to reach new unoccupied habitats, but colonization success also depends on biotic (e.g. species composition) and abiotic (e.g. temperature, moisture) factors. While weak within-species competition can initially favour establishment, low individual density also means limited mate availability. Consequently, the ability to self-fertilize is expected to increase colonization ability [1], and indeed self-compatibility and selfing are often associated with weedy habit [2], invasiveness [3,4] and ruderal strategies [5]. In pairwise comparisons, selfing species also tend to have larger species range than their outcrossing relatives [6]. However, range expansion and colonization of new habitats can also be costly. First, a colonization/competition trade-off is expected such that a good colonizer can be a poor competitor [7], thereby limiting range colonization to sufficiently open and disturbed habitats. Second, the specific demographic dynamics associated with range expansion, namely, recurrent bottlenecks followed by rapid population growth, can lead to the establishment of initially rare mutations, so-called 'allele surfing' [8]. This can contribute to the accumulation of deleterious mutations on the expansion front, creating an 'expansion load' [9–11].

Although these are clear theoretical predictions for the relationships between genomic and phenotypic characteristics of expanding populations,

those relationships have rarely been characterized in natural populations. Direct characterization of fitness and accumulation of mutations during range expansion have only been performed in experimentally evolving populations of bacteria [12] or in North American populations of *Arabidopsis lyrata* that have recently gone through a range expansion [11]. In *A. lyrata*, only overall individual plant performance was measured so it remains unclear whether and how different parts of the life cycle and individual competitive ability were affected during range expansion.

Shepherd's purse, *Capsella bursa-pastoris* (L.) Medik. (Brassicaceae) (*Cbp*), a species with a very large distribution due to a recent range expansion [13], is a good model species to test the relationship between the evolution of the life cycle and the accumulation of deleterious mutations during range expansion. It is an annual, selfing and allotetraploid species originating from the hybridization, about 100–300 thousand years ago, of two diploid species, *Capsella grandiflora* (Klokov) and *Capsella orientalis* (Fauché & Chaub.) Boiss [14,15]. *Cbp* probably first occurred in the Middle East (ME), then spread to Europe (EU) before invading eastern Asia (AS) and more recently spread worldwide as a result of human migrations [13]. Owing to limited gene flow between these geographic areas, three main genetic clusters can be delineated from both nuclear DNA and gene expression data [13,15,16]. Furthermore, as expected given the species history, the East Asian populations (i.e. colonization front) presented a reduced genetic diversity and a higher amount of deleterious mutations than the European and Middle Eastern ones (core populations) [15].

Selfing and polyploidization can confer an immediate advantage for the colonization of new habitats. Both mechanisms could explain the colonization and invasion successes of plant species: self-fertilization provides reproductive insurance, a single individual being able to initiate an invasion [5,17–19], while polyploidy allows partial sheltering from the negative effect of selfing, especially by masking deleterious alleles [20,21]. Two previous studies have started to investigate the differences in competitive ability of *Cbp* from its two parental species (*C. grandiflora* and *C. orientalis*) and a relative species, *Capsella rubella*, and within *Cbp* taking into account numerous populations across the species range [22,23]. They showed that (i) self-fertilizing species of *Capsella* are more sensitive to competition than outcrossing ones [22,23] and (ii) competitive ability decreased during range expansion in *Cbp* [23]. However, these studies did not consider all stages and fitness components of the life cycle and did not directly test for a possible effect of the load accumulated during range expansion.

Here, we conducted an experiment under controlled environmental conditions following the whole life cycle of *Cbp* accessions whose number of potentially deleterious mutations had been estimated from whole genome sequence data [15]. Life-history traits, fertility, viability of the progeny, phenological traits (germination time, flowering start) and vegetative traits (growth of the rosette) were assessed in *Cbp* accessions coming from the three main genetic clusters. In contrast with most studies based on one or a few life-history traits, we recorded the impact of different densities of competitors on traits associated with the complete life cycle, i.e. from the mother plant to the progeny. This allowed us to measure both fertility and viability, two major components of fitness, and to highlight the presence of trade-off between traits. We show that competitor density impacted only the fertility of the mother plant but not its viability. More generally, populations from the colonization front have a lower fitness than populations from the core of the distribution. However, contrary to our expectations, populations from the colonization front, which have a high genetic load, are the least affected by competition in the conditions of the experiment. That could be due to their early initiation of flowering, which allows them to partially avoid the effect of competition. These results also highlight the need to take the whole life cycle into account when characterizing the relationship between genetic load, population demography and the evolution of the phenotype.

## 2. Material and Methods

### (a) Plant sampling and seed preparation

#### (i) Sampling

We assessed lifetime fitness of accessions of *Cbp* from different parts of the natural range against different numbers of competitors. Following previous studies [23], we chose an annual species growing in similar environments, *Matricaria chamomilla* (Asteraceae), as competitor. The two species have similar distribution areas (Europe and temperate Asia) and co-occur, in particular in northern Greece where *M. chamomilla* appears as one of the main competitors of *Capsella* species (S. Glémin and M. Lascoux 2017, personal observations). Commercial seeds of *M. chamomilla* were used to ensure good germination and homogeneity among plants. We used 24 accessions of *Cbp*. Each accession corresponds to seeds harvested from a single maternal plant. The 24 accessions originated from different sampling sites that belong to one of the three main genetic clusters, Europe (EU), the Middle East (ME) and Asia (AS) [13]. More specifically, nine accessions came from sites distributed across Europe, 11 from sites in Asia, three from sites in the Middle East and one from the United States (figure 1*a*; electronic supplementary material, table S1). The latter belongs to the Middle East genetic cluster [16] and was thus added to the ME cluster in following analyses.

#### (ii) Seed preparation

For each accession, at least 30 seeds were surface-sterilized and sown into agar plates (electronic supplementary material, table S2). Following sterilization, seeds in agar plates were kept for 7 days at 4°C and in complete darkness for stratification. Agar plates were then transferred into a growth chamber (12 : 12 h light : darkness cycles, 22°C) for germination (approximately 2 days later). After one week, seedlings with well-developed cotyledons and a radicle at least 1 cm long were used for the competition experiment (approx. 20 seedlings per accession).

### (b) Competition experiment

#### (i) Experimental set-up

To assess the competitive abilities of the different *Cbp* accessions, we measured different life-history traits with one, two, four or eight competitors (*M. chamomilla*) or without (negative control) (electronic supplementary material, figure S1).

Seeds from accessions from Asia, Europe and the Middle East were put in agar plates (7 days with 24 h darkness at 4°C for stratification; then approximately two weeks at 12 : 12 h light : darkness, 22°C) and the germination rate was recorded ($GR_{MP}$, where MP stands for 'mother plant'). Only accessions with a high enough germination rate ($GR_{MP} > 0.3$) were used (electronic supplementary material, table S2).

Seedlings were transplanted into square pots (11 × 11 cm) and maintained in a growth chamber (12 : 12 h light : darkness cycles,

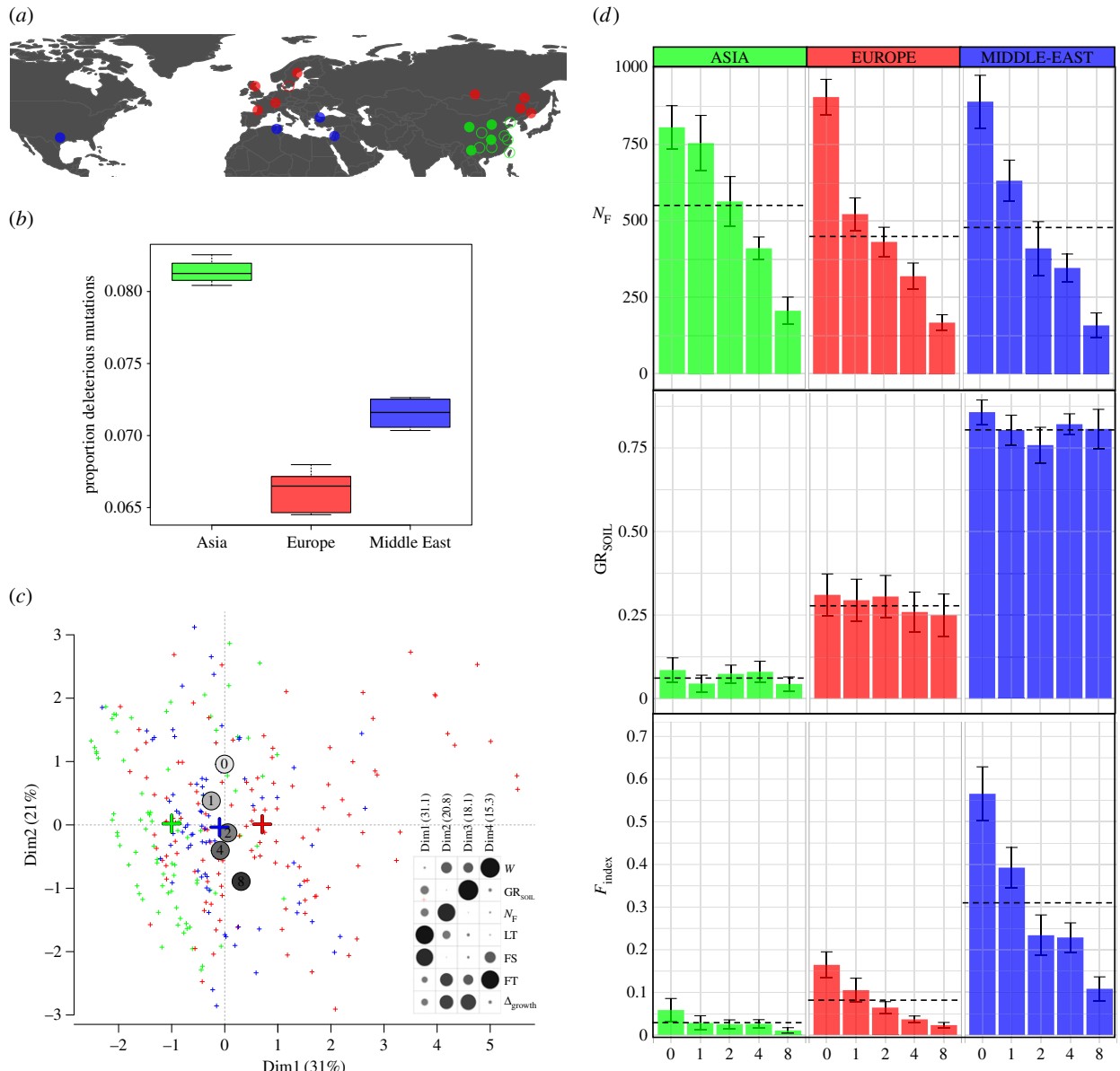

**Figure 1.** (*a*) Distribution of *Capsella bursa-pastoris* (*Cbp*) accessions used in the present study. Accessions belong to three main genetic clusters (Asia in green, Europe in red and Middle East in blue); empty symbol means that the germination rate was too low to include the accession in the experiment. (*b*) Proportion of deleterious mutations of *Cbp*, for all the accessions from Asia, Middle East and Europe. (*c*) Principal component analysis (PCA). Crosses correspond to the three main genetic clusters (Asia in green (left cross), Europe in red (right cross) and Middle East in blue (central cross)) while circles with numbers correspond to the number of competitors. The inset in highlights the variables that contribute the most to the four first dimensions of the PCA. The size of the circles is proportional to the contribution. (*d*) Number of flowers ($N_F$), germination rate of offspring seeds ($GR_{SOIL}$) and fitness index ($F_{index}$), which combines fertility and viability, for the five competitive environments and for the three main genetic clusters. The dashed lines correspond to the mean by genetical clusters (whatever the number of competitors). (Online version in colour.)

22°C). This pot size maximized competition while still permitting the focal plant development [22]. The competitors (*M. chamomilla*) were transplanted 3 days before *Cbp* accessions because of their slightly slower development (data not shown). Then, one *Cbp* individual was transplanted into the centre of the pot, with an average distance between the focal plant and competitor(s) of approximately 4 cm. For each accession, all treatments were replicated over four randomized blocks.

### (ii) *Capsella bursa-pastoris* performances

To assess viability and fertility under different competition intensities, phenotypic traits related to vegetative growth, phenology and fitness were recorded (electronic supplementary material, figure S1). First, the diameter of the rosette, i.e. the largest distance between opposite leaves, of each focal plant (in cm) was measured at 14 ($Dt_1$) and 21 ($Dt_2$) days after transplantation. The measurement was taken from pictures, using the program

ImageJ [24]. Rosette growth rate ($\Delta_{growth}$) was then computed as $\Delta_{growth} = Dt_2 - Dt_1$. Second, we recorded three phenological traits: (i) *flowering start* (FS), i.e. the date of the first flower appearance, (ii) *flowering timespan* (FT), i.e. the number of days between FS and plant death, and (iii) *lifetime* (LT), i.e. the number of days from plant transplantation to death. Finally, two major components of fitness were assessed: (i) *fertility*, $N_f$, estimated as the number of fruits produced, and (ii) *germination rate of the progeny*, tested either in soil ($GR_{SOIL}$) or in agar ($GR_{AGAR}$).

After the focal plant had dried out, up to 200 seeds were collected to investigate the fertility of the accession and the viability of its progeny. First, the *mean seed weight* (W), a good predictor of seed germination, was quantified as the weight of pooled seeds divided by their number (*ca* 200 seeds). Then, 25 randomly sampled seeds were germinated on agar plates ($GR_{AGAR}$; favourable environment) and 50 others were sown directly in soil of square pots (7 × 7 cm) to create a more 'natural' and stressful environment. After 7 days of stratification (24 h darkness, 4°C),

Petri dishes and pots were moved into growth chambers (12 : 12 h light : darkness, 22°C). We studied the *germination dynamic* (GD) by recording the germination rate after 1, 2 and 7 days for the seeds germinated in agar plates ($GD_{AGAR}$) and after 2, 5, 7 and 14 days for seeds sown in pots ($GD_{SOIL}$). The number of seed germinated divided by the number of seeds sown on the last day of each treatment was considered as the progeny *germination rate* ($GR_{AGAR}$ and $GR_{SOIL}$). Then, for each accession and three of the competition treatments (no competitor, two competitors and eight competitors), one seedling was transplanted from the agar plate into individual pots. The seedlings were grown until they started flowering ($FS_{prog}$).

## (c) Data analysis

### (i) Genetic load estimation

Kryvokhyzha *et al.* [15,25] estimated genetic load between populations of *Cbp* by classifying mutations into tolerated and deleterious ones using SIFT4G [26]. Briefly, SIFT4G classifies mutations according to conservation scores based on sequence alignments including a reference species and related species. Both *A. thaliana* and *C. rubella* reference partition databases were used to classify the single nucleotide polymorphisms (SNPs). To get only the annotation of the mutations that occurred after speciation of *Cbp*, the mutations with reconstructed ancestral sequences were polarized as in Kryvokhyzha *et al.* [15] and only derived mutations were analysed. This was done for the two subgenomes of *Cbp* the one derived from *C. grandiflora* and the one derived from *C. orientalis*, leading to two estimates of the proportion of deleterious mutations, $\mu_{delCg}$ and $\mu_{delCo}$, respectively. We then estimated the genetic load for each accession as $\mu_{del} = (\mu_{delCo} + \mu_{delCg})/2$ (figure 1*b*; electronic supplementary material, table S1).

### (ii) Principal component analysis

To obtain a synthetic view of the relationship between the three main clusters, with respect to fitness components, all the traits were jointly analysed with a principal component analysis (PCA, implemented in 'FactoMineR', R package v. 2.41-3 [27]). We thereafter focused on the first four principal components ('corrplot', R package v. 2.41-3 [28]), which captured most of the variation (85%).

### (iii) Germination

Cox's hazard models (*coxph* function, *survival* package v. 2.41-3; R software v. 3.3.1 [29,30]) were used to assess the effect of geographic origin ($G_{OR}$, a three-level factor, AS, EU and ME) on progeny germination dynamics ($GD_{AGAR}$ and $GD_{SOIL}$).

### (iv) Other traits

For the phenotypic traits (vegetative- and fitness-related), linear mixed models and generalized linear mixed models (LMM and GLMM, lme4 package v. 1.1-21, R [31]) were adjusted to the data to assess the effect of geographic origin, number of competitors and their interaction, while controlling for block effect:

$$\gamma_{ijkl} = C_j + G_k + (C_j \times G_k) + b_i + a_l + \varepsilon_{ijk}. \tag{2.1}$$

For a given trait, $\gamma$ is the observation for an accession *l* from geographic origin *k*, in block *i*, and with *j* competitors. Upper and lower case letters denote fixed and random effects, respectively. *C* is the competition-specific fixed effect, *G* is the geographic origin fixed effect (a three-level factor corresponding to the AS, EU and ME genetic clusters), ($C_j \times G_k$) is the interaction between competition and geographic origin, and $\varepsilon$ is the error term (table 1). The block effect (*b*; four-level factor) and the accession within each geographic origin (*a*; four from ME, four from AS and eight from EU) were included as random effects.

**Table 1.** Analyses of deviance. $GR_{MP}$: germination rate of mother plants, $\Delta_{growth}$: vegetative growth rate, LT: lifetime, FS: flowering start, FT: flowering time, $N_F$: number of fruits, W: seed weight, $GR_{SOIL}$, progeny germination rate in pots, $FS_{prog}$: flowering start of the progeny, $F_{index}$: fitness index, $I_c$: competition index. Degrees of freedom were 3, 2, 1 and 2, respectively, for block effect, geographic origin ($G_{OR}$), number of competitors ($N_c$) and the interaction term between $G_{OR}$ and $N_c$. The distributions of errors ($\varepsilon$) are: B, binomial; G, gamma; N, normal; NB, negative binomial. Significance levels are: ***$p < 0.001$; **$p < 0.01$; *$p < 0.05$; n.s.$p > 0.05$.

| trait | | $G_{OR}$ | $N_c$ | $G_{OR} \times N_c$ | $\varepsilon$ |
|---|---|---|---|---|---|
| $GR_{MP}$ | $\chi^2$ | 164*** | — | — | B |
| $\Delta_{growth}$ | F | 9.2* | 4.9* | 3.2 n.s. | N |
| LT | $\chi^2$ | 9.1* | 0.3 n.s. | 7.3* | NB |
| FS | $\chi^2$ | 9.34** | 14.28*** | 14.32*** | N |
| FT | $\chi^2$ | 3.7 n.s. | 13.7*** | 9.6** | NB |
| $N_F$ | $\chi^2$ | 2.2 n.s. | 225*** | 2.2 n.s. | NB |
| W | F | 20.3*** | 2.6 n.s. | 1.1 n.s. | N |
| $GR_{SOIL}$ | $\chi^2$ | 20*** | 10** | 1.4 n.s. | B |
| $FS_{prog}$ | $\chi^2$ | 14.93*** | 0.06 n.s. | 0.53 n.s. | N |
| $F_{index}$ | $\chi^2$ | 13.64** | 47.70*** | 6.77* | N |
| $I_c$ | $\chi^2$ | 36.8*** | 137*** | 0.32 n.s. | G |

When the interaction term between the geographic origin and the number of competitors was significant, sub-models, modelling either the effect of the geographic origin or the effect of the number of competitors, were used to estimate the effect of each factor independently as recommended by Crawley [32,33].

When the geographic origin had a significant effect on a given trait, the correlation between the trait and the average proportion of deleterious mutations between subgenomes ($\mu_{del}$) was estimated. As accessions from Asia had larger values of $\mu_{del}$ than accessions from Europe and the Middle East, we reported Spearman's $\rho$, which is more conservative than Pearson's product moment correlation, *r*, if outliers are present (both Spearman's $\rho$ and Pearson's *r* are reported in electronic supplementary material, table S3). Finally, between-trait correlations were estimated to investigate phenotypic trade-offs (electronic supplementary material, figure S5).

### (v) Competition index

To characterize the influence of competition intensity on fitness, we defined a competition index ($I_c$) based on the number of fruits produced ($N_F$) [22]. For a given accession *k*, and *i* competitors ($i = 1, 2, 4$ or 8), $I_c$ was computed as

$$I_{cki} = \frac{N_{F_{ki}}}{\bar{N}_{F_{k0}}}, \tag{2.2}$$

where the number of fruits for each plant ($N_{F_{ki}}$) was standardized by the average across the four replicates of the number of fruits in absence of competition ($\bar{N}_{F_{k0}}$); the lower the $I_c$, the higher the effect of competition. Significance of differences in response to competition for the various geographic origins was then assessed through GLMM with a gamma error term. Finally, for a given accession *k*, the average competitive index was used as an indicator of its sensitivity to competition.

### (vi) Fitness index

To consider the joint effect of fertility ($N_F$) and viability ($GR_{SOIL}$), we defined a fitness index ($F_{INDEX}$) as

$$F_{INDEX} = GR_{SOIL} \times N_F. \tag{2.3}$$

We investigated differences in relative fitness, which was log-transformed ($x + 1$), as a function of the number of competitors and the geographic origin using the same model as in equation (2.1). To facilitate reading, this index is scaled to vary between 0 and 1 ($F_{INDEX} = GR_{SOIL} \times N_F / \max(GR_{SOIL} \times N_F)$), meaning that the higher the index, the higher the relative fitness is.

### (vii) Model choice

To assess the significance of the fixed factors, models were simplified by removing interaction terms. The significance of difference in deviance explained between two models was assessed through a likelihood ratio test (LRT). Significance of difference between factor levels was tested through the same procedure and factor levels that were not significantly different were merged ($p > 0.05$) [33].

## 3. Results

Overall, both competition intensity and geographical area of origin strongly influenced phenotypic variation (figure 1c; electronic supplementary material, figure S2b). The first two principal components (PCs) aloone captured 52% of the total variance, and up to 85% of it was explained by PC1–PC4. Figure 1c shows that geographical origin discriminates individuals along PC1 (31% of total variance), while competition discriminates them along PC2 (21% of total variance). This orthogonality suggests that competition and geographic origin influenced different life-history traits. *Flowering start* (FS) and *lifetime* (LT) contributed most to PC1 and thus to differentiation of genetic clusters, while *fertility* ($N_F$) and, to a lesser extent, *flowering time* (FT) and *growth rate* ($\Delta_{growth}$) contributed most to PC2, reflecting the effect of competition intensity on these traits (figure 1c; electronic supplementary material, figure S2a).

For clarity, we will first focus on the effect of geographic origin on each life-history trait and then present the effects of competition. The effects of each parameter of our model (equation (2.1)) are summarized in table 1.

### (a) Reduced overall performance but earlier flowering of accessions from the colonization front

#### (i) Mother plants

*Germination.* Germination rate ($GR_{MP}$) varied greatly among accessions, ranging from 0 to 1. Five accessions belonging to the Asian cluster and two belonging to the European cluster were not used in the competition experiment because of their low germination rate ($GR_{MP} < 0.3$, electronic supplementary material, table S2). Middle Eastern accessions had the highest germination rate (electronic supplementary material, table S2). Across all accessions, $GR_{MP}$ was negatively correlated with the proportion of deleterious mutations ($\rho = -0.49$; electronic supplementary material, table S3).

*Vegetative growth.* Growth varied depending on geographic origin (table 1): European and Asian accessions grew at similar rates, but Middle East accessions showed a much faster growth rate (electronic supplementary material, table S4 and Figure S3b). However, difference in growth rate did not correlate with genetic load ($\rho = 0.04$, electronic supplementary material, table S3 and figure S6).

*Phenology. Lifetime* differed between the three geographic origins (table 1): Asian accessions had the shortest *lifetime* and European the longest (table 1; electronic supplementary material, table S4 and figure S3e). Because the interaction term between geographic origin ($G_{OR}$) and competitor ($N_C$) was significant for *flowering timespan* and *flowering start* (table 1), sub-models considering only the $G_{OR}$ were also computed. Asian accessions flowered first (electronic supplementary material, figure S3c) and had a longer *flowering timespan* than Middle Eastern and European accessions (table 1; electronic supplementary material, figure S3d). In addition, genetic load and phenology traits were strongly correlated. Accessions with the highest genetic load had the shortest *lifetime* but also the earliest *flowering start* (respectively, $\rho = -0.42$ and $\rho = -0.55$; electronic supplementary material, table S3 and figures S5 and S6); they also tended to have a longer *flowering timespan* but the correlation was much weaker ($\rho = 0.12$, $p = 0.04$). Finally, since *flowering start* did not affect *flowering timespan* (GLM, $t = 1.4$, d.f. = 283, $p = 0.16$), differences in *lifetime* were mainly due to differences in the length of the vegetative period.

*Reproductive traits.* Since the number of flowers and the number of flowering stems were strongly correlated with the total number of fruits ($r = 0.99$ and 0.8, respectively), only the total number of fruits ($N_F$) was used as a proxy for fertility. Though Asian accessions produced the highest number of fruits, fertility did not vary significantly among geographic origins (table 1).

#### (ii) Progeny

*Seed weight.* Asian accessions produced lighter seeds than European and Middle Eastern accessions (table 1; electronic supplementary material, table S4 and figure S3f), whose seed weight was similar. Thus, accessions with the highest proportion of deleterious mutations produced the lightest seeds ($\rho = -0.23$, electronic supplementary material, table S3 and figure S6)

*Germination rate.* Results for $GR_{AGAR}$ and $GR_{SOIL}$ were similar although, as expected, the differences between factors were amplified in the soil environment; only results for $GR_{SOIL}$ are reported in the main text (see electronic supplementary material, table S5 for $GR_{AGAR}$). For $GR_{SOIL}$, the interaction term between geographic origin ($G_{OR}$) and competitors ($N_C$) was significant (table 1) and a sub-model considering only the block effect and $G_{OR}$ was thus also used. Similar to mother plants, the geographic origin explained a large part of the variance in germination rate ($p < 0.001$). Seeds from Asian accessions had by far the lowest germination rate (only 6.5%), while 28 and 81% of the seeds from Europe and the Middle East, respectively, germinated (figure 2; electronic supplementary material, figure S3g). Interestingly, $GR_{SOIL}$ was correlated with seed weight ($\rho = 0.21$, electronic supplementary material, figure S5), i.e. the heavier the seeds, the higher the germination rate; in *Cbp*, seed weight is thus a good proxy of seed viability.

*Progeny phenology.* Asian accessions germinated significantly later than both European and Middle Eastern accessions (Cox's Hazard Model, all $p < 0.001$; figure 2; electronic supplementary material, table S5). However, as for mother plants, progeny ($FS_{prog}$) of Asian accessions started to flower earlier than Middle Eastern and European accessions (table 2; electronic supplementary material, table S4) despite their slightly later germination time. Hence Asian accessions reached maturity much faster than accessions from Europe or the Middle East. Interestingly, this trait is much more conserved between mother plants and offspring in Asian accessions ($r = 0.81$) than in European ($r = 0.46$) or Middle Eastern accessions ($r = 0.09$).

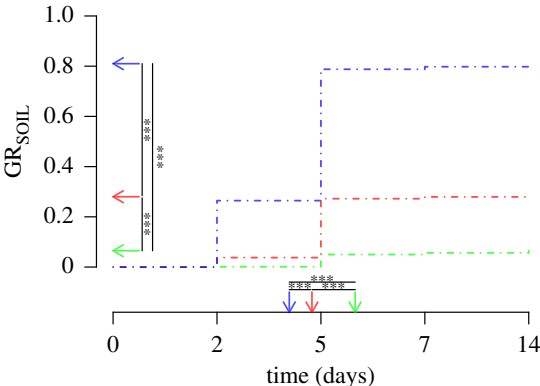

**Figure 2.** Germination rate of the progeny (GR$_{SOIL}$) in soil at different time points (days) according to geographical origin. The dot-dashed lines indicate the dynamic of the germination for the three geographical areas: Asia (green, GR = 0.07), Europe (red, GR = 0.28), Middle East (blue, GR = 0.81). The vertical arrows indicate the mean germination time, and the horizontal arrows the proportion of germination for each geographical area. The asterisks indicate significant differences between the three regions (*$p < 0.05$; **$p < 0.01$; ***$p < 0.001$). (Online version in colour.)

### (b) Unexpectedly, Asian accessions were the least sensitive to competition

#### (i) Life-history traits

The number of competitors ($N_C$) affected negatively many life-history traits (figure 1). First, overall *rosette growth rates* tended to decrease when $N_C$ increased, though it should be pointed out that this effect was mainly driven by a strong decrease in rosette size in the presence of eight competitors (table 1; electronic supplementary material, figure S4b). *Flowering timespan* and *flowering start* were the two phenology-related traits affected by the number of competitors (table 1); FT decreased as competition intensity increased ($t = -3.6$, $p < 0.001$; electronic supplementary material, figure S3d) and FS did not depend on competitor number for Asian ($t = -0.92$, $p = 0.36$) and was marginally significant for Middle Eastern ($t = 1.815$, $p = 0.07$) and significant for European accessions ($t = 4.1$, $p < 0.001$), meaning that flowering starts later under increased competition. While *fertility* (number of fruits, $N_F$) did not vary significantly between geographic areas, it was strongly affected by competition (table 1): the larger the number of competitors, the lower the number of fruits (electronic supplementary material, figure S3a). Because fertility is one of the main components of fitness, the ability of *Cbp* to withstand competition should strongly determine its fitness. Finally, it is interesting to note that competition intensity did not only affect the mother plant performances but also progeny establishment success as the germination rate of progeny (GR$_{SOIL}$) decreased when the mother plant had to face strong competition ($z = -2.17$, $p = 0.03$; table 1).

#### (ii) Competitive indices

As expected, the *competitive index* ($I_C$) decreased with increasing levels of competition, but more importantly the magnitude of the effect varied between geographic origins (table 1 and figure 3a). Surprisingly, whatever the number of competitors considered, Asian accessions had higher $I_C$ than Middle Eastern ($p = 0.001$) or European accessions ($p < 0.001$). In striking contrast with our expectation, the correlation between $I_C$ and genetic load was positive ($\rho = 0.67$, $p = 0.006$, figure 3b). This unexpected pattern is explained by the capacity of Asian accessions to

bloom early, thereby avoiding competition ($\rho = -0.79$, $p < 0.001$, figure 3c). To test this hypothesis, a generalized linear model (GLM) with *genetic load* and *flowering start* as explanatory variables was adjusted to the $I_C$ data. The relationship between *genetic load* and $I_C$ became non-significant ($\chi^2 = 3.2$, $p = 0.07$), while the negative relationship between *flowering start* and *competitive index* remained highly significant ($\chi^2 = 12.5$, $p < 0.001$).

#### (iii) Asian accessions show a reduction in overall fitness

$F_{index}$ was developed to integrate the two more important fitness components involved directly in survival and reproduction. $F_{index}$ clearly shows that Asian accessions generally performed more poorly than European, which in turn were worse than Middle Eastern ones (table 2 and figure 1d). A significant interaction between $G_{OR}$ and $N_c$ was due to the fact that Asian accessions were not sensitive to the increase in the number of competitors ($t = -1.33$, $p = 0.18$), while in European and Middle Eastern clusters, the $F_{index}$ decreased when the competitor number increased (EU: $t = -5.41$, $p < 0.001$; ME: $t = -6.35$, $p < 0.001$).

## 4. Discussion

In the present study, we analysed how accessions of *C. bursapastoris* from parts of the natural range with different demographic histories performed under increasing density of competitors. First, we showed that populations from the colonization front (Asia), which had the highest genetic load, also had a lower fitness than the populations of Europe and the Middle East. For all accessions, both vegetative growth and reproductive output were negatively affected by competitor density. However, somewhat unexpectedly given their higher genetic load and their lower absolute performances, Asian accessions were less affected by competition than European and Middle Eastern ones. A potential explanation could simply be that Asian accessions live fast and die young, which would limit the competition for resource acquisition before flowering. A shorter growing period before flowering would explain both the lower absolute performance and the weaker sensitivity to competition. Hence, a shift of ecological strategy in Asian accessions would allow them to cope with competition by avoiding it and could explain their successful establishment despite their higher genetic load. Finally, our study demonstrates the crucial importance of measuring multiple components of plant fitness and that analysing distant proxies of fitness can lead to misleading conclusions.

### (a) Front populations show reduced fitness consistent with their high genetic load

Deleterious mutations can impact both reproductive rate and juvenile competitive ability [9,34]. Our study validates this expectation since in most cases, the higher the genetic load, the lower the performance (electronic supplementary material, figure S6). However, in Asian accessions, the effect of the genetic load may have been partly mitigated by trade-offs between reproductive traits. The low germination rate of Asian accessions was indeed partly compensated by the number of fruits produced, which tended to be higher than for accessions from the core of the range (AS, 544; ME, 475; EU, 492; electronic supplementary material, figure S5). Asian seeds were also lighter than European and Middle Eastern ones and likely

Table 2. Summary of differences in key life-history traits across the three geographical clusters. Mean (±s.e.) for each trait is given by genetic cluster (AS = Asian accessions; EU = European accessions, ME = Middle Eastern accessions). The results are summarized per life-history trait (main trends) and the main conclusions are described.

| stage | life-history trait | | AS | EU | ME | main trend | main conclusion |
|---|---|---|---|---|---|---|---|
| focal plant | germination rate | $GR_{MP}$ | 0.29 ± 0.1 | 0.61 ± 0.1 | 0.77 ± 0.1 | AS < EU < ME | |
| | fertility (number of fruits) | $N_F$ | 544 ± 38 | 462 ± 31 | 486 ± 41 | AS = EU = ME | AS tended to have more fruits than EU and ME |
| | growth rate (cm day⁻¹) | $\triangle_{growth}$ | 0.85 ± 0.05 | 0.87 ± 0.03 | 1.2 ± 0.07 | AS = EU < ME | |
| | lifetime (days) | LT | 53 ± 1 | 70 ± 2 | 57 ± 1 | AS < ME < EU | AS had a shorter lifetime but flowered longer than ME and EU |
| | flowering timespan (days) | FT | 29 ± 1 | 27 ± 1 | 25 ± 1 | AS > ME = EU | |
| | flowering start (days) | FS | 23 ± 1 | 43 ± 1 | 31 ± 1 | AS < ME < EU | |
| progeny | germination rate | $GR_{SOIL}$ | 0.07 ± 0.01 | 0.28 ± 0.03 | 0.81 ± 0.02 | AS < EU < ME | the lighter the seeds, the lower the germination rate |
| | seed weight (μg seed⁻¹) | W | 87 ± 1.9 | 96 ± 1.4 | 97 ± 1 | AS < EU = ME | |
| | germination time (days) | $GD_{SOIL}$ | 6 + 0.19 | 5 + 0.03 | 4 + 0.03 | AS < ME = EU | AS reached reproductive stage much faster despite their late germination |
| | flowering start (days) | $FS_{prog}$ | 23 ± 1 | 36 + 0.7 | 31 ± 0.5 | AS < ME < EU | |
| fitness | relative fitness index | $F_{index}$ | 0.03 ± 0.01 | 0.08 ± 0.01 | 0.31 ± 0.03 | AS < EU < ME | owing to a weak germination rate, the global performance of AS was worse than EU which was worse than ME |
| competitiveness | competitive index | $I_c$ | 0.63 ± 0.06 | 0.35 ± 0.02 | 0.43 ± 0.04 | AS > ME = EU | AS were less affected by competitors than ME and EU accessions |

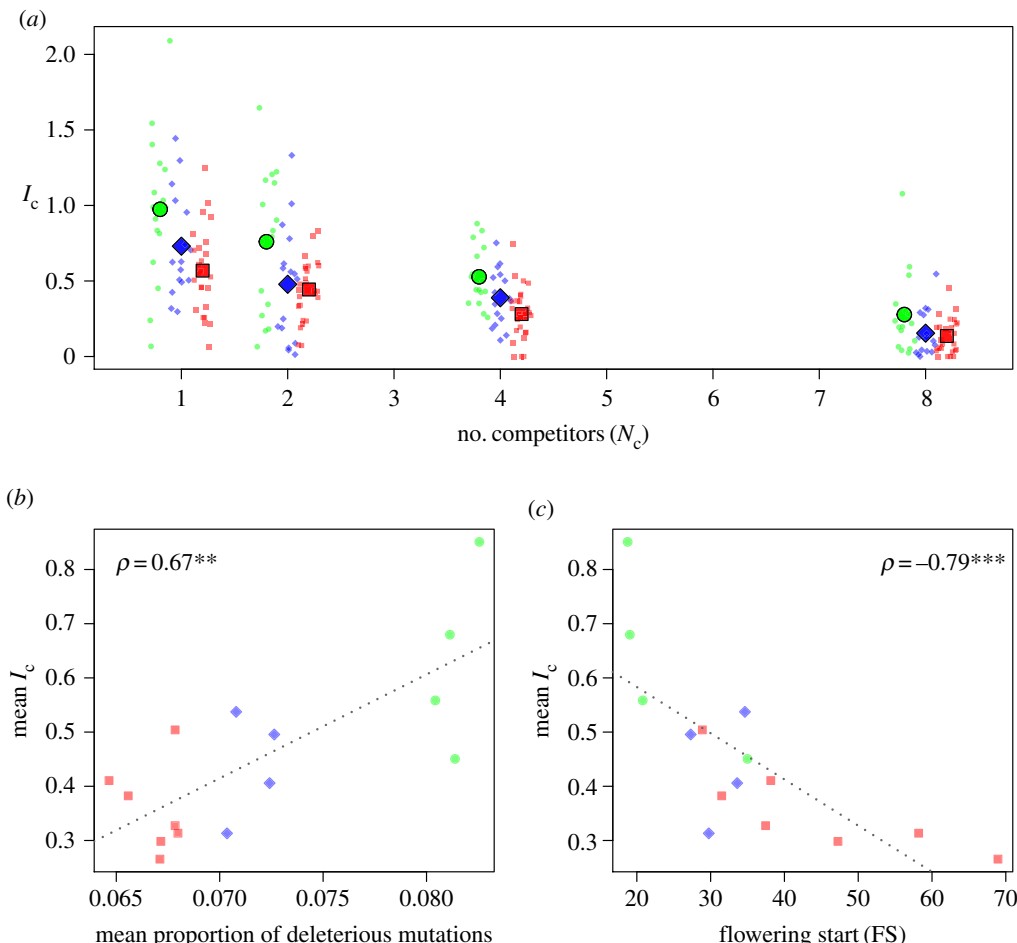

**Figure 3.** Sensitivity of *Capsella bursa-pastoris* to competition. (*a*) Competitive index ($I_c$) as a function of the number of competitors ($N_c$) for Asian accessions, green circles; Middle East accessions, blue diamonds; and European accessions, red squares. Large symbols show the mean $I_c$ for each geographic origin. Low values of $I_c$ indicate high competition. (*b*) Accession sensitivity to competition (average $I_c$ across the four replicates for each competitor number) as a function of the proportion of deleterious mutations or as a function of the *flowering start*. (*c*) For both relationships, Spearman's $\rho$ correlation coefficients are indicated as well as their significance levels: \*\*, $p < 0.01$; \*\*\*, $p < 0.001$. (Online version in colour.)

had a small endosperm and a lack of nutrient reserves. Since resources are limited, a trade-off between number of offspring and size of the progeny (and thus viability) is often observed. For instance, in flowering plants, Karrenberg & Suter [35] showed that reduced seed number was offset by increased seed mass and survival, which was reinforced by seed survival. Such a trade-off between size and survival of the progeny was also demonstrated in other organisms (e.g. birds and fish) [36,37]. In addition, a general trade-off between offspring number and offspring quality may be expected, in particular in populations on the expansion front, since a decrease in seed size could allow better dispersal [38]. Finally, it should be noted that since the number of accessions per geographical area is limited, the variance of genetic load within each geographic area is small, and teasing apart the effect of genetic load from that of the geographical area is challenging as these two effects are confounded. This difficulty could be alleviated by carrying out a new experiment with a higher number of accessions from a single geographic area, for instance Europe, where accessions exhibit the highest variance in genetic load.

## (b) Phenological shifts in front populations reduce the effect of competition

The reduced mean fitness we observed for Asian accessions could also be offset by their enhanced competitive abilities.

In flowering plants, phenology plays a major role in establishment success in new environments and/or on colonization fronts [39,40]. More specifically, flowering time is an important component of plant ecology because it plays a crucial role in community assembly [41,42]. A priority effect, i.e. invasive species bloom earlier than native ones, may contribute to plant community structure by limiting the impact on reproductive success of interspecific competition for pollinators, nutrient resources or light accessibility [40]. Our results support the hypothesis of priority effect, the accessions from the colonization front flowering earlier than those from the core of the range. This phenological shift coupled with the rapid development of Asian accessions gave them easier access to resources (light and nutrients) and thereby limited the impact of competitors. As a result, the Asian accessions suffer less from competition than the European and Middle Eastern ones.

A shift in phenology can also be due to a high degree of phenotypic plasticity or represent an adaptive response to different environmental conditions [43–45] (e.g. day length, temperature, hygrometry, frost damage, pollinators). An adaptive response could explain why, in general, the Asian accessions responded differently from accessions from the two other genetic clusters (figure 1), and in particular, the fact that the shift of phenology is also observed in the absence of competitor. Further experiments with competitors installed

at different times would allow us to understand if the lower sensitivity of Asian accessions to competition is only due to a shift in phenology or if some other traits, for example, a more developed root system permitting a better access to nutrient resources underground, are also involved.

However, there are some important caveats. First, our experiment considered only one set of environmental conditions and we cannot exclude that our results would have been different using a different set of parameters. However, a common garden experiment carried out in three localities (Sweden, 59° N; Canada, 43° N; China, 23° N) with different climatic conditions showed the same pattern, i.e. AS flowering before ME and EU and the same absolute lower performance of Asian accessions [46], suggesting that our results are rather consistent. Second, in contrast with our study, Yang et al. [23] found a lower competitive ability of Asian accessions. However, this could be due to differences in experimental set-up between the two studies; in Yang et al. [23], Capsella individuals were sown at the same time as competitors, not with a 3 day delay as in our experiment. The latter has likely exacerbated the differences in phenology between accessions and favoured Asian accessions which flowered first. Moreover, we discarded accessions with very low germination rates, which may also have biased the experiment towards the fittest Asian accessions, compared with Yang et al. [23], who used a wider range of accessions. Third, we have so far discussed ecological explanations for the difference in response to competition between Asian accessions and accessions from the core of the range. However, other, non-ecological, factors may also contribute. First, the shift in phenology of the Asian accessions may simply be a direct consequence of their high genetic load and may not represent an ecological adaptation [47,48]. Indeed, the genetic load could be responsible for changes in life-history traits and, for example, individuals with a higher load could simply use resources less efficiently and thereby have a lower vegetative growth rate, which could in turn lead to an early reproductive period. Second, introgression from a locally adapted species, C. orientalis, in the Asian populations of Cbp may also have influenced phenology and facilitated range expansion [15]. In some flowering plants, introgression with native species could facilitate invasion by an alien species; for instance, hybrids resulting from crossing and introgression between Carpobrotus edulis and its native congener Carpobrotus chilensis are known to be successful invaders of coastal plant communities [49,50]. Another example concerns the annual sunflower, Helianthus annuus, which has been able to expand its ecological amplitude and geographic range by introgression with locally adapted native species [51]. The introgression of chromosomal segments from a locally adapted species may thus also have facilitated the expansion of Cbp in Asia in spite of its higher genetic load.

## (c) The whole life cycle matters when assessing fitness

Finally, numerous studies have detected the presence of negative effects of competition on plant growth and reproduction [52,53], but no study, to our knowledge, considered the impact on progeny, i.e. on the complete life cycle. As phenotypic trade-offs can occur between different life stages, it is of paramount importance to assess fitness from life-history traits recorded during the whole life cycle. However, the number of flowers, fruits or seeds produced is too often considered as the sole proxy of fitness. Our study demonstrates that the presence of complex trade-offs between traits at different times ing the life cycle can completely change the interpretation of the results. Competition affected not only fruit production but also seed quality. Thus, although Asian accessions tended to produce more fruits (which could have led to a higher fitness), competition on the mother plant also reduced the germination rate of its progeny and thereby decreased strongly its fitness. Fully understanding the consequences of competition on plants, and in our case its relationships with accumulation of deleterious mutations, thus requires a thorough characterization of plant performances across the whole life cycle.

Data accessibility. The datasets, containing the raw data for each measured trait, are available from the Dryad Digital Repository: https://doi.org/10.5061/dryad.vt4b8gtp7 [54].

Authors' contributions. M.O., S.G. and M.L. designed the experiment; M.O., J.H. and J.G. collected data; M.O., J.H. and P.M. analysed output data; M.O., P.M., S.G. and M.L. wrote the first draft of the manuscript, and all authors contributed substantially to revisions.

Competing interests. We declare we have no competing interests.

Funding. This study was supported by the Erik Philip Sörensen Foundation and the Swedish Research Council.

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
