## [Reviewer comments · Proceedings of the Royal Society B: Biological Sciences]

Review History

RSPB-2019-1707.R0 (Original submission)

Review form: Reviewer 1

Recommendation

Major revision is needed (please make suggestions in comments)

Scientific importance: Is the manuscript an original and important contribution to its field?

Good

General interest: Is the paper of sufficient general interest?

Good

Quality of the paper: Is the overall quality of the paper suitable?

Acceptable

Is the length of the paper justified?

Yes

Should the paper be seen by a specialist statistical reviewer?

No

Do you have any concerns about statistical analyses in this paper? If so, please specify them explicitly in your report.

Yes

It is a condition of publication that authors make their supporting data, code and materials available - either as supplementary material or hosted in an external repository. Please rate, if applicable, the supporting data on the following criteria.

Is it accessible?

Yes

Is it clear?

No

Is it adequate?

No

Do you have any ethical concerns with this paper?

No

Comments to the Author

Major comments

The authors present an interesting common garden experiment to test whether populations from the invasion front differ in phenology and competitive ability to those from the natural distribution. They find very interesting evidence that although fitness is overall lower at the front, these plants possessed better competitive ability.

While the manuscript is well written and the data seem appropriately analysed, I felt as though the results could be better described and that there were some nuances in the data that the authors could highlight to better test the question they are answering. More specifically,

1) It seems strange that the authors analysed all traits univariately, and then presented a multivariate PCA. Is there a justification for not continuing the multivariate analysis to compare differences in competition and geography? Relevant to this, the authors cite phenotypic trade-offs as responsible for the observed patterns (e.g., L.35, L.381-382), but do not present any correlations among the traits to quantify these trade-offs. If correlations among traits differ between geography or competition (as the PCA suggests), this would be very interesting and support their conclusions very well.

2) The principal components analysis is very interesting, it seems as though the effects of geography are orthogonal to the effects of competition. This would suggest that competition acts upon different traits compared to expanding to new geographic areas. I think this needs to be highlighted, and tested specifically using a two-way MANOVA for example (there is a natural groupings in the data so PCA may not be the best method, and doesn't give a statistical test). On this note, Fig.1 was also difficult to interpret (especially without a legend), is it possible to use colours for region, and then different symbols for competition?

3) Noting from Fig.1, I was wondering if the authors checked for differences in variance? It seems as though European accessions have higher variance in both PC1 and PC2, which would be interesting because the invasion front could be associated with reduced phenotypic variance due to strong selection, inbreeding or selfing. This also contrasts with Fig.2, where Asian accessions showed higher variation in response to competition, suggesting that inbreeding/drift could

determine expansion potential. I feel as though these could be results as well.

4) The use of likelihood ratio tests seems strange (e.g., L245-L249 & L278-284). Are these univariate tests removing the variable of interest? Why not just report the results of the ANOVA and linear model summary with post-hoc pairwise comparisons?

5) I was a little disappointed not to see the proportional hazards curves presented (Table 2). These could help to visualise how the regions differed.

I also had some concerns over how natural this experiment was, considering another study (Yang et al. 2018) seemed to have done the same test and find different results. Here, the authors conclude this is due a difference in methodology (L.422-429). If competition changes depending on whether seeds are planted at the same time versus three days later, then this has important and has significant biological meaning. This needs to be better justified.

Minor comments

There is some vague language at the start of the introduction. For example, the use of 'ecological processes' (L.42) does not say much, when something like 'suitable habitat' is more descriptive and accurate. Same for 'genetic factors' (L.40). Also, on L.53 'alternatively' doesn't follow from the previous argument. I think this first paragraph should be split and written more clearly.

L.131 Could you tell the reader how many seedlings per accession on average?

L.189 I found the equation difficult to follow. It would be better if the authors used more simpler letter notation to denote each term. For example, $C_j \times G_k$ for the competition-by-geographic origin.

L.191 & L.214 The use of experimental block as a fixed, rather than random effect should be justified. It makes me wonder if they were avoiding random effects models.

L.197 & L.258 What are 'sub-models'? This should be described better

L.210 To make this equation more accessible to the readers, authors could describe it more simply as the number of fruits standardised by the mean in the absence of competition (or something similar).

L.286 'weighted' should be 'weighed'

Review form: Reviewer 2

Recommendation

Major revision is needed (please make suggestions in comments)

Scientific importance: Is the manuscript an original and important contribution to its field?

Good

General interest: Is the paper of sufficient general interest?

Excellent

Quality of the paper: Is the overall quality of the paper suitable?

Good

Is the length of the paper justified?

Yes

Should the paper be seen by a specialist statistical reviewer?

No

Do you have any concerns about statistical analyses in this paper? If so, please specify them explicitly in your report.

No

It is a condition of publication that authors make their supporting data, code and materials available - either as supplementary material or hosted in an external repository. Please rate, if applicable, the supporting data on the following criteria.

Is it accessible?

No

Is it clear?

No

Is it adequate?

N/A

Do you have any ethical concerns with this paper?

No

Comments to the Author

Summary: The authors' manuscript, "Ecological strategy and genetic load in the shepherd's purse (*Capsella bursa-pastoris*) from the core and the limit of its natural range", presents an interesting and extensive study on the interaction of competition with genetic load and fitness proxies across the species range of an autotetraploid throughout its lifespan. I commend the authors for what must have been an immense amount of work in the greenhouse with the multi-generational, lifespan measures and data collection that went into this study. I think this is great work that offers valuable insights to the field. I have several comments/questions/suggestions for understanding and improving the manuscript.

Major points:

1. I think a major point (if not the main major point) the authors try to make at a broader scale in this study is that it is necessary to measure different fitness traits throughout lifespan to get an accurate picture of 'true' fitness (long-term reproductive success). If I am correct in my assessment, then I am confused as to why the competitive index only takes into account the number of fruits produced. I see that this has a historical definition, but can what the authors deem to be a better measure of fitness (offspring germination success, or some combination of traits?) be used as a comparison competitive index. We could then see if this does indeed change the overall story in terms of whether simply counting fruits is sufficient or not to represent fitness, or if a novel competitive index that accounts for multiple important fitness-related traits is similar enough that future studies would suffice to only count fruits.
2. I think the genetic load estimates play a major role in this paper (the title includes mention of genetic load), but I found the methods description of this aspect to be very short and entirely require going to the previous study that is cited. I think sufficient explanation should exist within this manuscript for readers to understand exactly what was done to obtain the load measures used in this paper. For example, the equation in lines 177-178 uses the terms del_Co and del_Cg which are not mentioned elsewhere in this paper and therefore confusing. Table S1 which is cited as well seems to instead be a list of sample locations, so I am not sure what was meant to be referred to here?

I lack clarity in the main text as to whether all accessions used in this study are the same as those

in the study cited for deleterious load proportions might be needed. I was initially confused in line 177 where it says load was the average proportion, which I thought meant across accessions, but this is across sub-genomes, so it should be correct that load is on a per-accession basis? And deleterious sites are identified using SIFT4G on nonsynonymous mutations in each accession? I would mention the use of SIFT on these sites in the main text here.

3. There are a lot of results in this study, and I find it difficult to digest all of them. Along these lines I have several comments:

3a. In lines 236-240, how is GR_MP different from GR_AGAR and GR_SOIL? Line 146 is the first mention of GR_MP with no definition, and then it next appears here in line 236, but I do not see what it is nor what MP stands for? Is the negative correlation of GR_MP with deleterious load only in Middle-Eastern accessions, or is it across all accession? It could perhaps be nice to see these split into region (Table S3 is all accessions not divided into region), but see my next comment.

3b. I am not sure how much I learn from figure 1, in case there are figure limits, but I did think it might be useful to have a multi-panel figure showing each trait measured (germination rate, growth rate, ...) as compared to the genetic load per individual accession. Accessions could then be colored by region and significant correlations could be highlighted by, e.g., thicker regression lines on the plot. Number of competitors could be indicated by point size, but then I begin to add more confusing levels... I am imagining something along the lines of Figure 3 in Bosshard et al 2017, Genetics which would let the readers immediately see an overall view of what traits correlate and in which directions. I imagine this might allow for a significant reduction in the text for the results, or at the very least a clearer overall picture of the direction of correlations for ecological traits with load across the accessions (and thus across geographic space as well). From Table S3 I think this could be very neat because it looks like the majority of significant correlations are negative, except for flowering time and growth, which could make sense if very loaded individuals "live fast and die young" in order to have any reproductive success at all.

4. I do not see it as a fatal flaw, but I found the flow of the discussion difficult to follow when trying to keep all of the results in mind. I am not sure I have a good suggestion for improvement, so would just ask the authors to at least take a moment to ponder any potential improvement. For example line 431 begins with discussing a shift in phenology, but it is difficult to follow from the previous paragraph as that ends with several sentences referring to another study. Even just mentioning "A shift in phenology to earlier flowering at the range edge can be due to..." would have helped, as I had to go back and reread the previous paragraph in its entirety. If I continue in the paragraph starting at 431, it is a similar structure to others where the result is discussed, a future direction is mentioned, a caveat is mentioned, and a previous study is mentioned. Perhaps it is personal preference, but I think I am less prone to get lost if all the caveats across all results appear in a paragraph along with the other relevant previous studies that may make them not caveats / not relevant for the study at hand. Then when I am reading the parts of the discussion interpreting the results, it is simply interpretation and implications for the future.

5. Following up on my first major comment, I think the last paragraph of the discussion is again a major take-home from the paper. The discussion might be better served by putting this first. I would clarify the writing in this paragraph to be clear that Asian accessions have the most genetic load, yet the highest fitness measured by fruit count – a contradictory result, until you look at your other fitness measures (seed quality or others?) that show the lowest inferred fitness (thus also go back to my major comment 2 in relation to this).

6. Do any caveats about genetic load in a tetraploid need to be mentioned? Maybe this was discussed in the previous paper. But maybe masking of deleterious recessive variants due to the larger genome size could be a relevant phenomena that I do not think is able to be addressed in this study. Unless the authors also want to re-do analyses with the proportions in each sub-genome independently, but I think a simple mention of this caveat may be sufficient.

Minor points:

1. I would add a legend of the point colors directly on the plot in Figure 2 rather than necessitating reading into the caption.

2. In Figure 2B and 2C, I take μ in the y-axis to indicate mean, but it is not immediately stated in the caption. On the x-axis of 2B, I think it should not be a μ , but is rather a proportion? It would be one less step for readers' thinking if 'mean' and 'mean proportion' were written out (I think there is sufficient space).

Review form: Reviewer 3 (James Buckley)

Recommendation

Major revision is needed (please make suggestions in comments)

Scientific importance: Is the manuscript an original and important contribution to its field?

Good

General interest: Is the paper of sufficient general interest?

Excellent

Quality of the paper: Is the overall quality of the paper suitable?

Good

Is the length of the paper justified?

Yes

Should the paper be seen by a specialist statistical reviewer?

No

Do you have any concerns about statistical analyses in this paper? If so, please specify them explicitly in your report.

Yes

It is a condition of publication that authors make their supporting data, code and materials available - either as supplementary material or hosted in an external repository. Please rate, if applicable, the supporting data on the following criteria.

Is it accessible?

No

Is it clear?

No

Is it adequate?

No

Do you have any ethical concerns with this paper?

No

Comments to the Author

In this manuscript, Orsucci et al. use accessions sampled from three geographic regions with differing colonization history to assess phenotypic evolution during the recent invasion of *Capsella bursa-pastoris* into Asia. Their data suggest that populations from the more recently colonized part of the range have evolved distinct ecological strategies through changes in several fitness-related traits. Furthermore, they show that the earlier flowering phenology of recently colonised Asian populations makes them less susceptible to interspecific competition than accessions from other regions. In my opinion, the most novel aspect of this work is the focus on

tradeoffs among traits representing the full life cycle and their ability to connect this to competitive ability. The described associations between the phenotypic data and estimates of genetic load are interesting, but, as they note, do not disentangle the confounding effects of genetic load and geographic area (lines 372-377). Nevertheless, this work presents an interesting attempt to combine ecological, phenotypic and genetic data to better understand the processes underlying recent range expansions. Generally, the paper is well-written and the main findings clearly highlighted, though I have a number of questions and comments about certain aspects of the manuscript. My main comments are as follows:

1) I found it difficult to identify which results support the statement that “accessions from the front performed worse for most life history traits than those from the core” (e.g. lines 30-31, 101-102, 347-348). For example, progeny germination rate was clearly poorer for accessions from recently colonized populations, but fruit production is clearly higher for these same accessions. It is also not clear to me that variation in other traits (such as lifespan) can be classified as ‘better’ or ‘worse’. To better highlight this key finding, I would therefore suggest the following changes: (a) more clearly highlight in the results section those traits most relevant for assessing fitness e.g. fruit production and progeny viability; (b) Replace figure 1 (PCA) with a modified version of Figure S1. Instead of the photos, I would include two plots highlighting variation in the most relevant life history traits (fruit production and progeny viability). To illustrate effects of competition, you could select a single competition level to compare to your control treatment in these plots. (c) Where possible, provide relevant means and standard errors in a table form (including sample sizes), rather than the text. I understand that space is limited, but you could then delete this information from the results, making this section easier to read.

2) Given the aim of the paper to link genomic and phenotypic characteristics of the accessions, I do not understand why so many levels of competition were used. Would it not have been better to focus on one level of competition (e.g. 4 competitors vs no competition) and use more replicates per accession (or accessions per geographic area) to better estimate fitness-related traits? Could you please justify this experimental design in the introduction and methods?

3) I have several questions about the statistical analyses performed. (a) Given accession was nested within geographic area, how in your analyses did you account for pseudoreplication? I think generalised linear mixed effects models (GLMMs) would be more appropriate in this case than the GLM you use (lines 185-195). GLMMs would allow you to include the random effects of both ‘block’ and ‘accession nested within geographic area’. (b) Lines 221-222: I am not familiar with this approach of merging factor levels to simplify models. If I understand correctly, this is quite different to the standard methods of comparing factor levels using posthoc comparisons or defining contrasts in the context of more complex GLMs. Can you justify why you adopted the approach you did?

Minor comments and suggested edits:

Line 42: expand this to “ecological and evolutionary processes.”

Lines 53-57: The topic of these sentences seems slightly disconnected from the rest of the paragraph. I think you should end the first paragraph at line 53 with a statement about the knowledge gap your study aims to test. Then you can start a new paragraph providing some more background on competition and range expansion.

Lines 58-63: Again, this paragraph seems out-of-place and interrupts the flow of the introduction. Could you instead mention the benefits of allopolyploidisation later when you introduce your study system?

Line 88: Could you add an additional sentence describing the work conducted in the reference [24] – such information would be useful to better understand the novelty of the findings in the

present manuscript.

Line 152-3: "The growth of rosette size..." should be replaced by "Rosette growth rates..."

Line 170-172: In the results you refer to data on the start of flowering for progeny, but you do not mention the collection of this data in the methods. Can you add these details? Furthermore, given the progeny plants started flowering, why did you not additionally collect data on progeny fruit set?

Line 225-231: I would remove this summary, as the PCA is not easy to interpret (see comment above) and the approach is not described in the methods. You could keep the sentence beginning "For clarity...", as I agree that this is a useful way of presenting the results.

Can you present the results for associations with genetic load in one paragraph at the end of the results section? You could use a subtitle like "Associations between genetic load and fitness-related traits", and this would make it easier to compare the different traits. If you have space, you could also consider moving part of Table S3 (with Spearman's rank results) into the main text. Then, you could also remove the associated statistics from this written section.

I think it is clearer to avoid describing methods in the results section. Could you remove or move the following sentences to the methods section (e.g. lines 240-242; 278-279; 286-288)?

Line 300: "Complex phenology" is a confusing term - I would instead label this as "Progeny germination rates and reproductive phenology"

Figure 2: Can you overlay standard deviations (or boxplots) onto the plots? It is difficult to assess the differences among geographic regions with just means.

Table 2 is not that informative - can it be moved to the supplementary and replaced with Figure S4?

Line 379: "...the lowest vegetative growth and reproductive output..." - But the Asian accessions had the highest reproductive output as assessed by fruit production. Be careful with your phrasing here.

Lines 395-408: I would remove this section, as it does not seem relevant to the main topic for discussion - it is simply a (nice) overview of the transition from outcrossing to selfing.

Line 412-413: Can you provide references to support this statement?

Line 450-455: Are there any references to support this line of argument? I am not sure I understand how the "shift in phenology... may simply be a direct consequence of high genetic load". If there are no references, I would rephrase this line of argument.

Line 466-478: I agree that studying the whole life cycle can be particularly informative and is one of the key strengths of this study.

Supplementary files:

Figure S3: Why are not all combinations of competitor treatments and traits presented here? For example, there is no plot providing fruit number for each geographic area. It might be useful. Can you provide these plots here or explain why they are not included in the figure legend?

Decision letter (RSPB-2019-1707.R0)

28-Oct-2019

Dear Miss Orsucci:

I am writing to inform you that your manuscript RSPB-2019-1707 entitled "Ecological strategy and genetic load in the shepherd's purse (*Capsella bursa-pastoris*) from the core and the limit of its natural range" has, in its current form, been rejected for publication in Proceedings B.

This action has been taken on the advice of referees, who have recommended that substantial revisions are necessary. With this in mind we would be happy to consider a resubmission, provided the comments of the referees are fully addressed. However please note that this is not a provisional acceptance.

Sincerely,
Dr Sasha Dall
<mailto:proceedingsb@royalsociety.org>

Associate Editor
Board Member: 1
Comments to Author:

All reviewers agree that this is an interesting finding, of general interest. I also agree that integrating evolutionary responses with estimates of genetic load and core/edge comparisons is an important step forward.

However, all three reviewers have important concerns with the analysis, and some aspects of its validity, including concerns about pseudoreplication in the GLM models. Many of these concerns demand a reanalysis of much of the data.

The reviewers also make important suggestions that would improve the clarity of the figures and tables, to make interpretation clearer, assuming that your key findings are still supported following reanalysis.

For these reasons, I am rejecting this MS in its current state, with the possibility of a resubmission that fully tackles the reviewers' points, or fully justifies the statistical approaches taken.

Reviewer(s)' Comments to Author:

Referee: 1

Comments to the Author(s)

Major comments

The authors present an interesting common garden experiment to test whether populations from the invasion front differ in phenology and competitive ability to those from the natural distribution. They find very interesting evidence that although fitness is overall lower at the front, these plants possessed better competitive ability.

While the manuscript is well written and the data seem appropriately analysed, I felt as though the results could be better described and that there were some nuances in the data that the authors could highlight to better test the question they are answering. More specifically,

1) It seems strange that the authors analysed all traits univariately, and then presented a multivariate PCA. Is there a justification for not continuing the multivariate analysis to compare differences in competition and geography? Relevant to this, the authors cite phenotypic trade-offs as responsible for the observed patterns (e.g., L.35, L.381-382), but do not present any correlations among the traits to quantify these trade-offs. If correlations among traits differ between geography or competition (as the PCA suggests), this would be very interesting and support their conclusions very well.

2) The principal components analysis is very interesting, it seems as though the effects of geography are orthogonal to the effects of competition. This would suggest that competition acts upon different traits compared to expanding to new geographic areas. I think this needs to be highlighted, and tested specifically using a two-way MANOVA for example (there is a natural groupings in the data so PCA may not be the best method, and doesn't give a statistical test). On this note, Fig.1 was also difficult to interpret (especially without a legend), is it possible to use colours for region, and then different symbols for competition?

3) Noting from Fig.1, I was wondering if the authors checked for differences in variance? It seems as though European accessions have higher variance in both PC1 and PC2, which would be interesting because the invasion front could be associated with reduced phenotypic variance due to strong selection, inbreeding or selfing. This also contrasts with Fig.2, where Asian accessions showed higher variation in response to competition, suggesting that inbreeding/drift could determine expansion potential. I feel as though these could be results as well.

4) The use of likelihood ratio tests seems strange (e.g., L245-L249 & L278-284). Are these univariate tests removing the variable of interest? Why not just report the results of the ANOVA and linear model summary with post-hoc pairwise comparisons?

5) I was a little disappointed not to see the proportional hazards curves presented (Table 2). These could help to visualise how the regions differed.

I also had some concerns over how natural this experiment was, considering another study (Yang et al. 2018) seemed to have done the same test and find different results. Here, the authors conclude this is due a difference in methodology (L.422-429). If competition changes depending on whether seeds are planted at the same time versus three days later, then this has important and has significant biological meaning. This needs to be better justified.

Minor comments

There is some vague language at the start of the introduction. For example, the use of 'ecological processes' (L.42) does not say much, when something like 'suitable habitat' is more descriptive and accurate. Same for 'genetic factors' (L.40). Also, on L.53 'alternatively' doesn't follow from the previous argument. I think this first paragraph should be split and written more clearly.

L.131 Could you tell the reader how many seedlings per accession on average?

L.189 I found the equation difficult to follow. It would be better if the authors used more simpler letter notation to denote each term. For example, $C_j \times G_k$ for the competition-by-geographic origin.

L.191 & L.214 The use of experimental block as a fixed, rather than random effect should be justified. It makes me wonder if they were avoiding random effects models.

L.197 & L.258 What are 'sub-models'? This should be described better

L.210 To make this equation more accessible to the readers, authors could describe it more simply as the number of fruits standardised by the mean in the absence of competition (or something similar).

L.286 'weighted' should be 'weighed'

Referee: 2

Comments to the Author(s)

Summary: The authors' manuscript, "Ecological strategy and genetic load in the shepherd's purse (*Capsella bursa-pastoris*) from the core and the limit of its natural range", presents an interesting and extensive study on the interaction of competition with genetic load and fitness proxies across the species range of an autotetraploid throughout its lifespan. I commend the authors for what must have been an immense amount of work in the greenhouse with the multi-generational, lifespan measures and data collection that went into this study. I think this is great work that offers valuable insights to the field. I have several comments/questions/suggestions for understanding and improving the manuscript.

Major points:

1. I think a major point (if not the main major point) the authors try to make at a broader scale in this study is that it is necessary to measure different fitness traits throughout lifespan to get an accurate picture of 'true' fitness (long-term reproductive success). If I am correct in my assessment, then I am confused as to why the competitive index only takes into account the number of fruits produced. I see that this has a historical definition, but can what the authors deem to be a better measure of fitness (offspring germination success, or some combination of traits?) be used as a comparison competitive index. We could then see if this does indeed change the overall story in terms of whether simply counting fruits is sufficient or not to represent fitness, or if a novel competitive index that accounts for multiple important fitness-related traits is similar enough that future studies would suffice to only count fruits.

2. I think the genetic load estimates play a major role in this paper (the title includes mention of genetic load), but I found the methods description of this aspect to be very short and entirely require going to the previous study that is cited. I think sufficient explanation should exist within this manuscript for readers to understand exactly what was done to obtain the load measures used in this paper. For example, the equation in lines 177-178 uses the terms del_Co and del_Cg which are not mentioned elsewhere in this paper and therefore confusing. Table S1 which is cited as well seems to instead be a list of sample locations, so I am not sure what was meant to be referred to here?

I lack clarity in the main text as to whether all accessions used in this study are the same as those in the study cited for deleterious load proportions might be needed. I was initially confused in line 177 where it says load was the average proportion, which I thought meant across accessions, but this is across sub-genomes, so it should be correct that load is on a per-accession basis? And deleterious sites are identified using SIFT4G on nonsynonymous mutations in each accession? I would mention the use of SIFT on these sites in the main text here.

3. There are a lot of results in this study, and I find it difficult to digest all of them. Along these lines I have several comments:

3a. In lines 236-240, how is GR_MP different from GR_AGAR and GR_SOIL? Line 146 is the first mention of GR_MP with no definition, and then it next appears here in line 236, but I do not see what it is nor what MP stands for? Is the negative correlation of GR_MP with deleterious load only in Middle-Eastern accessions, or is it across all accession? It could perhaps be nice to see these split into region (Table S3 is all accessions not divided into region), but see my next comment.

3b. I am not sure how much I learn from figure 1, in case there are figure limits, but I did think it might be useful to have a multi-panel figure showing each trait measured (germination rate, growth rate, ...) as compared to the genetic load per individual accession. Accessions could then be colored by region and significant correlations could be highlighted by, e.g., thicker regression lines on the plot. Number of competitors could be indicated by point size, but then I begin to add more confusing levels... I am imagining something along the lines of Figure 3 in Bosshard et al 2017, Genetics which would let the readers immediately see an overall view of what traits correlate and in which directions. I imagine this might allow for a significant reduction in the text for the results, or at the very least a clearer overall picture of the direction of correlations for ecological traits with load across the accessions (and thus across geographic space as well). From Table S3 I think this could be very neat because it looks like the majority of significant correlations are negative, except for flowering time and growth, which could make sense if very loaded individuals "live fast and die young" in order to have any reproductive success at all.

4. I do not see it as a fatal flaw, but I found the flow of the discussion difficult to follow when trying to keep all of the results in mind. I am not sure I have a good suggestion for improvement, so would just ask the authors to at least take a moment to ponder any potential improvement. For example line 431 begins with discussing a shift in phenology, but it is difficult to follow from the previous paragraph as that ends with several sentences referring to another study. Even just mentioning "A shift in phenology to earlier flowering at the range edge can be due to..." would have helped, as I had to go back and reread the previous paragraph in its entirety. If I continue in the paragraph starting at 431, it is a similar structure to others where the result is discussed, a future direction is mentioned, a caveat is mentioned, and a previous study is mentioned. Perhaps it is personal preference, but I think I am less prone to get lost if all the caveats across all results appear in a paragraph along with the other relevant previous studies that may make them not caveats / not relevant for the study at hand. Then when I am reading the parts of the discussion interpreting the results, it is simply interpretation and implications for the future.

5. Following up on my first major comment, I think the last paragraph of the discussion is again a major take-home from the paper. The discussion might be better served by putting this first. I would clarify the writing in this paragraph to be clear that Asian accessions have the most genetic load, yet the highest fitness measured by fruit count – a contradictory result, until you look at your other fitness measures (seed quality or others?) that show the lowest inferred fitness (thus also go back to my major comment 2 in relation to this).

6. Do any caveats about genetic load in a tetraploid need to be mentioned? Maybe this was discussed in the previous paper. But maybe masking of deleterious recessive variants due to the larger genome size could be a relevant phenomena that I do not think is able to be addressed in this study. Unless the authors also want to re-do analyses with the proportions in each sub-genome independently, but I think a simple mention of this caveat may be sufficient.

Minor points:

1. I would add a legend of the point colors directly on the plot in Figure 2 rather than necessitating reading into the caption.
2. In Figure 2B and 2C, I take μ in the y-axis to indicate mean, but it is not immediately stated in the caption. On the x-axis of 2B, I think it should not be a μ , but is rather a proportion? It would be one less step for readers' thinking if 'mean' and 'mean proportion' were written out (I think there is sufficient space).

Referee: 3

Comments to the Author(s)

In this manuscript, Orsucci et al. use accessions sampled from three geographic regions with differing colonization history to assess phenotypic evolution during the recent invasion of *Capsella bursa-pastoris* into Asia. Their data suggest that populations from the more recently colonized part of the range have evolved distinct ecological strategies through changes in several fitness-related traits. Furthermore, they show that the earlier flowering phenology of recently colonised Asian populations makes them less susceptible to interspecific competition than accessions from other regions. In my opinion, the most novel aspect of this work is the focus on tradeoffs among traits representing the full life cycle and their ability to connect this to competitive ability. The described associations between the phenotypic data and estimates of genetic load are interesting, but, as they note, do not disentangle the confounding effects of genetic load and geographic area (lines 372-377). Nevertheless, this work presents an interesting attempt to combine ecological, phenotypic and genetic data to better understand the processes underlying recent range expansions. Generally, the paper is well-written and the main findings clearly highlighted, though I have a number of question and comments about certain aspects of the manuscript. My main comments are as follows:

- 1) I found it difficult to identify which results support the statement that "accessions from the front performed worse for most life history traits than those from the core" (e.g. lines 30-31, 101-102, 347-348). For example, progeny germination rate was clearer poor for accessions from recently colonized populations, but fruit production is clearly higher for thee same accessions. It is also not clear to me that variation in other traits (such as lifespan) can be classified as 'better' or 'worse'. To better highlight this key finding, I would therefore suggest the following changes: (a) more clearly highlight in the results section those traits most relevant for assessing fitness e.g. fruit production and progeny viability; (b) Replace figure 1 (PCA) with a modified version of Figure S1. Instead of the photos, I would include two plots highlighting variation in the most relevant life history traits (fruit production and progeny viability). To illustrate effects of competition, you could select a single competition level to compare to your control treatment in these plots. (c) Where possible, provide relevant means and standard errors in a table form (including sample sizes), rather than the text. I understand that space is limited, but you could then delete this information from the results, making this section easier to read.
- 2) Given the aim of the paper to link genomic and phenotypic characteristics of the accessions, I do not understand why so many levels of competition were used. Would it not have been better to focus on one level of competition (e.g. 4 competitors vs no competition) and use more replicates per accession (or accessions per geographic area) to better estimate fitness-related traits? Could you please justify this experimental design in the introduction and methods?
- 3) I have several questions about the statistical analyses performed. (a) Given accession was nested within geographic area, how in your analyses did you account for pseudoreplication? I think generalised linear mixed effects models (GLMMs) would be more appropriate in this case than the GLM you use (lines 185-195). GLMMs would allow you to include the random effects of both 'block' and 'accession nested within geographic area'. (b) Lines 221-222: I am not familiar with this approach of merging factor levels to simplify models. If I understand correctly, this is quite different to the standard methods of comparing factor levels using posthoc comparisons or

defining contrasts in the context of more complex GLMs. Can you justify why you adopted the approach you did?

Minor comments and suggested edits:

Line 42: expand this to “ecological and evolutionary processes.”

Lines 53-57: The topic of these sentences seems slightly disconnected from the rest of the paragraph. I think you should end the first paragraph at line 53 with a statement about the knowledge gap your study aims to test. Then you can start a new paragraph providing some more background on competition and range expansion.

Lines 58-63: Again, this paragraph seems out-of-place and interrupts the flow of the introduction. Could you instead mention the benefits of allopolyploidisation later when you introduce your study system?

Line 88: Could you add an additional sentence describing the work conducted in the reference [24] – such information would be useful to better understand the novelty of the findings in the present manuscript.

Line 152-3: “The growth of rosette size...” should be replaced by “Rosette growth rates...”

Line 170-172: In the results you refer to data on the start of flowering for progeny, but you do not mention the collection of this data in the methods. Can you add these details? Furthermore, given the progeny plants started flowering, why did you not additionally collect data on progeny fruit set?

Line 225-231: I would remove this summary, as the PCA is not easy to interpret (see comment above) and the approach is not described in the methods. You could keep the sentence beginning “For clarity...”, as I agree that this is a useful way of presenting the results.

Can you present the results for associations with genetic load in one paragraph at the end of the results section? You could use a subtitle like “Associations between genetic load and fitness-related traits”, and this would make it easier to compare the different traits. If you have space, you could also consider moving part of Table S3 (with Spearman’s rank results) into the main text. Then, you could also remove the associated statistics from this written section.

I think it is clearer to avoid describing methods in the results section. Could you remove or move the following sentences to the methods section (e.g. lines 240-242; 278-279; 286-288)?

Line 300: “Complex phenology” is a confusing term – I would instead label this as “Progeny germination rates and reproductive phenology”

Figure 2: Can you overlay standard deviations (or boxplots) onto the plots? It is difficult to assess the differences among geographic regions with just means.

Table 2 is not that informative – can it be moved to the supplementary and replaced with Figure S4?

Line 379: “...the lowest vegetative growth and reproductive output...” – But the Asian accessions had the highest reproductive output as assessed by fruit production. Be careful with your phrasing here.

Lines 395-408: I would remove this section, as it does not seem relevant to the main topic for discussion – it is simply a (nice) overview of the transition from outcrossing to selfing.

Line 412-413: Can you provide references to support this statement?

Line 450-455: Are there any references to support this line of argument? I am not sure I understand how the “shift in phenology... may simply be a direct consequence of high genetic load”. If there are no references, I would rephrase this line of argument.

Line 466-478: I agree that studying the whole life cycle can be particularly informative and is one of the key strengths of this study.

Supplementary files:

Figure S3: Why are not all combinations of competitor treatments and traits presented here? For example, there is no plot providing fruit number for each geographic area. It might be useful. Can you provide these plots here or explain why they are not included in the figure legend?

Author's Response to Decision Letter for (RSPB-2019-1707.R0)

See Appendix A.

RSPB-2020-0463.R0

Review form: Reviewer 1

Recommendation

Accept with minor revision (please list in comments)

Scientific importance: Is the manuscript an original and important contribution to its field?

Good

General interest: Is the paper of sufficient general interest?

Good

Quality of the paper: Is the overall quality of the paper suitable?

Good

Is the length of the paper justified?

Yes

Should the paper be seen by a specialist statistical reviewer?

No

Do you have any concerns about statistical analyses in this paper? If so, please specify them explicitly in your report.

No

It is a condition of publication that authors make their supporting data, code and materials available - either as supplementary material or hosted in an external repository. Please rate, if applicable, the supporting data on the following criteria.

Is it accessible?

No

Is it clear?

No

Is it adequate?

No

Do you have any ethical concerns with this paper?

No

Comments to the Author

The authors present an interesting common garden experiment to test whether populations from the invasion front differ in phenology and competitive ability to those from the natural distribution. They find very interesting evidence that although fitness is overall lower at the front, these plants possessed better competitive ability.

Overall, I think the authors have done well to address the comments made in the previous review and I appreciate the level of detail in their response, especially in their adjustments to the discussion. I have no major criticisms, just some minor comments below.

1) The first two paragraphs do not frame the topic very clearly. For example, L.52-55, the use of 'peculiar demographic dynamics' is not required, and the authors mention "allele surfing" and "expansion load" without really defining what these mean. I think the introduction could read better if the first two paragraphs framed the topic a bit more clearly. The second paragraph only highlights what has been done before and does not make any statement about what is important to understand, for example, it could benefit from outlining the importance of understanding how genetic load and competition affects fitness at the expanding edge of the range.

2) Further to point 1, I think when describing the experiment, the authors could better highlight the hypothesis they are testing and what they predictions are. At the moment, the introduction is a little descriptive of the study system rather than focusing on the theory they are testing.

3) L.200 the use of vertical lines to denote random effect are not (in my opinion) a good idea as these mathematically signify absolute values. I suggest using upper case for fixed effects and lower case for random effects if the authors wish to clearly differentiate the two.

Review form: Reviewer 2

Recommendation

Accept as is

Scientific importance: Is the manuscript an original and important contribution to its field?

Excellent

General interest: Is the paper of sufficient general interest?

Excellent

Quality of the paper: Is the overall quality of the paper suitable?

Good

Is the length of the paper justified?

Yes

Should the paper be seen by a specialist statistical reviewer?

No

Do you have any concerns about statistical analyses in this paper? If so, please specify them explicitly in your report.

No

It is a condition of publication that authors make their supporting data, code and materials available - either as supplementary material or hosted in an external repository. Please rate, if applicable, the supporting data on the following criteria.

Is it accessible?

No

Is it clear?

No

Is it adequate?

No

Do you have any ethical concerns with this paper?

No

Comments to the Author

I apologize for my delay in returning this review; I was overly optimistic about my productivity levels during the coronavirus lockdown and unwisely accepted several manuscript reviews, putting this one later in the list.

After reading the revised version of the authors' manuscript as well as the comments in response to my previous reviews, I am happy to say I am satisfied with the responses and edits. I really think this is a great study, and am glad that it reads more clearly and cleanly now. I have no further comments or edits to request.

Review form: Reviewer 3 (James Buckley)

Recommendation

Accept with minor revision (please list in comments)

Scientific importance: Is the manuscript an original and important contribution to its field?

Excellent

General interest: Is the paper of sufficient general interest?

Excellent

Quality of the paper: Is the overall quality of the paper suitable?

Excellent

Is the length of the paper justified?

Yes

Should the paper be seen by a specialist statistical reviewer?

No

Do you have any concerns about statistical analyses in this paper? If so, please specify them explicitly in your report.

No

It is a condition of publication that authors make their supporting data, code and materials available - either as supplementary material or hosted in an external repository. Please rate, if applicable, the supporting data on the following criteria.

Is it accessible?

No

Is it clear?

N/A

Is it adequate?

N/A

Do you have any ethical concerns with this paper?

No

Comments to the Author

In this manuscript, Orsucci et al. use accessions sampled from three geographic regions with differing colonization history to assess phenotypic evolution during the recent invasion of *Capsella bursa-pastoris* into Asia. Their data suggest that populations from the more recently colonized part of the range in Asia have evolved a distinct ecological strategy. Specifically, the Asian populations show reduced overall fitness, consistent with the costs of an elevated genetic load, but a shift towards earlier flowering that makes them less susceptible to interspecific competition than accessions from other regions. The authors have made significant efforts to improve the manuscript based on previous reviewer comments, and I think the introduction and results are now much easier to digest. The new analyses with the mixed models are consistent with their original analyses, and the figures and tables in the main text now better highlight the main findings of the manuscript. I only have a number of minor comments that I think would help improve the clarity of the manuscript and particularly parts of the discussion.

Minor changes

Introduction, Methods and Results:

Line 33: I find the use of the phrase “performed worse for most life history traits” unclear, as it is not apparent what “performed worse” means. Given you produce a combined fitness index, I would simply state “showed reduced fitness” here and elsewhere in the manuscript (e.g. line 357).

Line 101: change to “the highest genetic load” or “a high genetic load”

Line 102: perhaps replace “very early flowering start” with “early initiation of flowering”

Line 259: you could be more specific with your subtitle here: “Reduced germination rates, but earlier flowering”.

Line 267-267: I would rephrase this sentence: “European or Asian accessions grew at similar rates, but Middle-East accessions showed a much faster growth rate”

Line 273: should be “competitors”

Line 278: replace “lowest lifetime” with “shortest lifetime”

Line 298: replace “as for mother plants” with “Similar to mother plants”

Line 332: change “when the competition increased” to “with increasing levels of competition”

Line 344: I would adjust this subheading to: “Asian accessions show reduced overall fitness”

Discussion, tables and figures:

Line 367: “...considering measures of fitness that are integrative and as close as possible to fitness itself ...” I find this a little unclear; perhaps you could simplify this to: “measuring multiple components of plant fitness...”

Line 371: This subtitle makes a strong statement given that you only assess correlations among traits and genetic load, and that the number of fruits produced did not significantly differ among geographic accessions. I would rephrase this to something like: “Recently-colonised populations show reduced fitness consistent with their higher genetic load”. Line 375 should read “may have been partly mitigated by...”

Line 381-382: I would suggest the following rephrasing: “... showed that reduced seed number was offset by increased seed mass and survival”

Line 393: this subtitle is not very descriptive – could it become a concluding statement? For example, “Phenological shifts in recently-colonised populations reduce effect of competition”

Line 394-395: Rephrase this sentence: “The reduced mean fitness we observed for Asian accessions could also be offset by their enhanced competitive abilities

Line 397-402: I don’t really see the relevance of discussing priority effects here. Unless I misunderstand the term, your data does not show that these early flowering accessions shape subsequent community assembly. I would therefore replace this section with a simple (re)statement of your finding that accessions from the colonisation front flowered earlier than those from the range core.

Lines 393-445: To break up this currently long section and make it easier to follow arguments, I would start new paragraphs at the following locations: line 405 (“A shift in phenology...” and 416 (“There are some important caveats”)? And 429 (“We have so far discussed ecological explanations”).

Line 405-407: The distinction between “high phenotypic plasticity” and “an adaptive response to different environmental conditions” is not immediately clear to me – could you expand on this a bit further?

Line 411-412: I am not sure I understand this sentence – perhaps remove?

Line 427: change to “very low germination rates”

Table 1: What does the final column refer to? I presume it is the error term, but it is best to explain that here in the table legend.

Table 2: I think this is a useful table to summarise the key results for the many traits measured, but I would rephrase the table title: “Summary of differences in key life history traits across the three geographical groups”. I would also change some of the column headings: replace “main trends” with “significant trends” and “additional comment” with “main conclusion”. Note that in the final column “than ME and EU” is repeated twice in one box.

The figure legends appear to be missing from the main text.

Decision letter (RSPB-2020-0463.R0)

11-Apr-2020

Dear Miss Orsucci

I am pleased to inform you that your manuscript RSPB-2020-0463 entitled "Ecological strategy and genetic load in the shepherd's purse (*Capsella bursa-pastoris*) from the core and the limit of its natural range" has been accepted for publication in Proceedings B.

The referee(s) have recommended publication, but also suggest some minor revisions to your manuscript. Therefore, I invite you to respond to the referee(s)' comments and revise your manuscript. Because the schedule for publication is very tight, it is a condition of publication that you submit the revised version of your manuscript within 7 days. If you do not think you will be able to meet this date please let us know.

- DNA sequences: Genbank accessions F234391-F234402

- Phylogenetic data: TreeBASE accession number S9123
- Final DNA sequence assembly uploaded as online supplemental material
- Climate data and MaxEnt input files: Dryad doi:10.5521/dryad.12311

[http://datadryad.org/submit?journalID=RSPB&manu=\(Document not available\)](http://datadryad.org/submit?journalID=RSPB&manu=(Document%20not%20available)) which will take you to your unique entry in the Dryad repository. If you have already submitted your data to dryad you can make any necessary revisions to your dataset by following the above link. Please see <https://royalsociety.org/journals/ethics-policies/data-sharing-mining/> for more details.

Sincerely,

Dr Sasha Dall
mailto:proceedingsb@royalsociety.org

Associate Editor
Board Member

Associate Editor
Board Member
Comments to Author:

Many thanks for your extensive revisions of the MS. All reviewers and myself agree re: the value and importance of your study. Please ensure that the outstanding issues are fully addressed (and explained) in your revised submission. In particular, one reviewer would like you to make clearer the hypotheses you are testing in your study, in order to frame your findings in a more general context. It would also - in my view - be useful to make your title more informative - in terms of what the key finding is of your study, and its broader relevance. Currently your title seems rather descriptive, and could perhaps be made more informative.

The reviewers also note that at the moment you have not uploaded or made accessible the data used for this study. This needs to be addressed before publication, please.

Reviewer(s)' Comments to Author:

Referee: 1

Comments to the Author(s).

The authors present an interesting common garden experiment to test whether populations from the invasion front differ in phenology and competitive ability to those from the natural distribution. They find very interesting evidence that although fitness is overall lower at the front, these plants possessed better competitive ability.

Overall, I think the authors have done well to address the comments made in the previous review and I appreciate the level of detail in their response, especially in their adjustments to the discussion. I have no major criticisms, just some minor comments below.

1) The first two paragraphs do not frame the topic very clearly. For example, L.52-55, the use of ‘peculiar demographic dynamics’ is not required, and the authors mention “allele surfing” and “expansion load” without really defining what these mean. I think the introduction could read better if the first two paragraphs framed the topic a bit more clearly. The second paragraph only highlights what has been done before and does not make any statement about what is important to understand, for example, it could benefit from outlining the importance of understanding how genetic load and competition affects fitness at the expanding edge of the range.

2) Further to point 1, I think when describing the experiment, the authors could better highlight the hypothesis they are testing and what they predictions are. At the moment, the introduction is a little descriptive of the study system rather than focusing on the theory they are testing.

3) L.200 the use of vertical lines to denote random effect are not (in my opinion) a good idea as these mathematically signify absolute values. I suggest using upper case for fixed effects and lower case for random effects if the authors wish to clearly differentiate the two.

Referee: 3

Comments to the Author(s).

In this manuscript, Orsucci et al. use accessions sampled from three geographic regions with differing colonization history to assess phenotypic evolution during the recent invasion of *Capsella bursa-pastoris* into Asia. Their data suggest that populations from the more recently colonized part of the range in Asia have evolved a distinct ecological strategy. Specifically, the Asian populations show reduced overall fitness, consistent with the costs of an elevated genetic load, but a shift towards earlier flowering that makes them less susceptible to interspecific competition than accessions from other regions. The authors have made significant efforts to improve the manuscript based on previous reviewer comments, and I think the introduction and results are now much easier to digest. The new analyses with the mixed models are consistent with their original analyses, and the figures and tables in the main text now better highlight the main findings of the manuscript. I only have a number of minor comments that I think would help improve the clarity of the manuscript and particularly parts of the discussion.

Minor changes

Introduction, Methods and Results:

Line 33: I find the use of the phrase “performed worse for most life history traits” unclear, as it is not apparent what “performed worse” means. Given you produce a combined fitness index, I would simply state “showed reduced fitness” here and elsewhere in the manuscript (e.g. line 357).

Line 101: change to “the highest genetic load” or “a high genetic load”

Line 102: perhaps replace “very early flowering start” with “early initiation of flowering”

Line 259: you could be more specific with your subtitle here: “Reduced germination rates, but earlier flowering”.

Line 267-267: I would rephrase this sentence: “European or Asian accessions grew at similar rates, but Middle-East accessions showed a much faster growth rate”

Line 273: should be “competitors”

Line 278: replace “lowest lifetime” with “shortest lifetime”

Line 298: replace “as for mother plants” with “Similar to mother plants”

Line 332: change “when the competition increased” to “with increasing levels of competition”

Line 344: I would adjust this subheading to: “Asian accessions show reduced overall fitness”

Discussion, tables and figures:

Line 367: "...considering measures of fitness that are integrative and as close as possible to fitness itself ..." I find this a little unclear; perhaps you could simplify this to: "measuring multiple components of plant fitness..."

Line 371: This subtitle makes a strong statement given that you only assess correlations among traits and genetic load, and that the number of fruits produced did not significantly differ among geographic accessions. I would rephrase this to something like: "Recently-colonised populations show reduced fitness consistent with their higher genetic load". Line 375 should read "may have been partly mitigated by..."

Line 381-382: I would suggest the following rephrasing: "... showed that reduced seed number was offset by increased seed mass and survival"

Line 393: this subtitle is not very descriptive – could it become a concluding statement? For example, "Phenological shifts in recently-colonised populations reduce effect of competition"

Line 394-395: Rephrase this sentence: "The reduced mean fitness we observed for Asian accessions could also be offset by their enhanced competitive abilities"

Line 397-402: I don't really see the relevance of discussing priority effects here. Unless I misunderstand the term, your data does not show that these early flowering accessions shape subsequent community assembly. I would therefore replace this section with a simple (re)statement of your finding that accessions from the colonisation front flowered earlier than those from the range core.

Lines 393-445: To break up this currently long section and make it easier to follow arguments, I would start new paragraphs at the following locations: line 405 ("A shift in phenology..." and 416 ("There are some important caveats")? And 429 ("We have so far discussed ecological explanations").

Line 405-407: The distinction between "high phenotypic plasticity" and "an adaptive response to different environmental conditions" is not immediately clear to me – could you expand on this a bit further?

Line 411-412: I am not sure I understand this sentence – perhaps remove?

Line 427: change to "very low germination rates"

Table 1: What does the final column refer to? I presume it is the error term, but it is best to explain that here in the table legend.

Table 2: I think this is a useful table to summarise the key results for the many traits measured, but I would rephrase the table title: "Summary of differences in key life history traits across the three geographical groups". I would also change some of the column headings: replace "main trends" with "significant trends" and "additional comment" with "main conclusion". Note that in the final column "than ME and EU" is repeated twice in one box.

The figure legends appear to be missing from the main text.

Referee: 2

Comments to the Author(s).

I apologize for my delay in returning this review; I was overly optimistic about my productivity levels during the coronavirus lockdown and unwisely accepted several manuscript reviews, putting this one later in the list.

After reading the revised version of the authors' manuscript as well as the comments in response to my previous reviews, I am happy to say I am satisfied with the responses and edits. I really think this is a great study, and am glad that it reads more clearly and cleanly now. I have no further comments or edits to request.

Author's Response to Decision Letter for (RSPB-2020-0463.R0)

See Appendix B.

Decision letter (RSPB-2020-0463.R1)

20-Apr-2020

Dear Miss Orsucci

I am pleased to inform you that your manuscript entitled "Shift in ecological strategy helps marginal populations of shepherd's purse (*Capsella bursa-pastoris*) to overcome a high genetic load" has been accepted for publication in Proceedings B.

Open Access

Paper charges

Sincerely,

Proceedings B
mailto: proceedingsb@royalsociety.org

Appendix A

Associate Editor

Board Member: 1

Comments to Author:

All reviewers agree that this is an interesting finding, of general interest. I also agree that integrating evolutionary responses with estimates of genetic load and core/edge comparisons is an important step forward.

However, all three reviewers have important concerns with the analysis, and some aspects of its validity, including concerns about pseudoreplication in the GLM models. Many of these concerns demand a reanalysis of much of the data.

The reviewers also make important suggestions that would improve the clarity of the figures and tables, to make interpretation clearer, assuming that your key findings are still supported following reanalysis.

For these reasons, I am rejecting this MS in its current state, with the possibility of a resubmission that fully tackles the reviewers' points, or fully justifies the statistical approaches taken.

Dear editor,

You will find below, in blue, the responses to the reviewers' comments on our manuscript. Their main concerns were the lack of clarity of some analyses, the statistical approach used and the fact that the result section was hard to follow because of too many results. We agree with the reviewers that these parts of the manuscript needed to be improved and we did our best to do so. More specifically, we have i) reanalyzed the dataset following the referees' suggestions using mixed models, ii) modified figures and tables included in the main text according to referees' recommendations and iii) edited the Results section in order to minimize the number of results presented while keeping the essential information. Finally we hope that the discussion now flows better. We have tried to accommodate the best we could, given the limited space available, the sometime divergent recommendations of the reviewers.

Regarding statistical analyses the following table presents, for each response variable (Traits) and each explanatory variable (Gor, geographic origin; Nc, number of competitor and their interaction Gor x Nc) the results using mixed model or not (as in previous version of the manuscript). Globally the conclusions drawn from the various models are the same (the changes are in bold cases and highlighted in gray).

Analyses of deviance

Traits	G_{OR}	G_{OR} - Mixed	N_C	N_C - Mixed	G_{OR} x N_C	G_{OR} x N_C - Mixed	ℰ
GR_{MP}	164 ^{***}	-	-	-	-	-	B
Δ_{growth}	14.4 ^{***}	9.2 [*]	4.9 [*]	4.9 [*]	1.3 ^{ns}	3.2 ^{ns}	N
LT	105 ^{***}	9.1 [*]	0.7 ^{ns}	0.3 ^{ns}	3.9 ^{ns}	7.3 [*]	NB
FS	77.5 ^{***}	9.3 ^{**}	2.3 ^{ns}	14.3 ^{***}	1.1 ^{ns}	14.3 ^{***}	N
FT	9.2 ^{**}	3.7 ^{ns}	13.4 ^{***}	13.7 ^{***}	9.9 ^{**}	9.6 ^{**}	NB
N_F	4.8 ^{ns}	2.2 ^{ns}	162 ^{***}	227.8 ^{***}	1 ^{ns}	2.1 ^{ns}	NB
W	20.3 ^{***}	7.8 [*]	2.6 ^{ns}	3.8 ^{ns}	1.1 ^{ns}	0.9 ^{ns}	N
GR_{SOIL}	5408 ^{***}	20 ^{***}	21.5 ^{***}	10 ^{**}	4.5 ^{ns}	1.4 ^{ns}	B
FS_{prog}	65.2 ^{***}	14.9 ^{***}	0.4 ^{ns}	0.1 ^{ns}	0.3 ^{ns}	0.5 ^{ns}	N
I_C	36.8 ^{***}	11.8 ^{**}	137 ^{***}	53 ^{***}	0.32 ^{ns}	0.25 ^{ns}	G

GR_{MP}: germination rate of mother plants, *Δ_{growth}*: vegetative growth rate, *LT*: life time, *FS*: flowering start, *FT*: flowering time, *N_F*: number of fruits, *W*: seed weight, *GR_{SOIL}*: progeny germination rate in pots, *FS_{prog}*: flowering start of the progeny; *I_C*: competition index). The degrees of freedom were 2, 1 and 2, respectively for geographic origin (*G_{OR}*), number of competitors (*N_C*) and the interaction term between *G_{OR}* and *N_C*. Significance levels are ^{***}, $p < 0.001$; ^{**}, $p < 0.01$; ^{*}, $p < 0.05$; ^{ns}, $p > 0.05$.

Reviewer(s)' Comments to Author:

Referee: 1

Comments to the Author(s)
Major comments

The authors present an interesting common garden experiment to test whether populations from the invasion front differ in phenology and competitive ability to those from the natural distribution. They find very interesting evidence that although fitness is overall lower at the front, these plants possessed better competitive ability.

Thanks.

While the manuscript is well written and the data seem appropriately analysed, I felt as though the results could be better described and that there were some nuances in the data that the authors could highlight to better test the question they are answering.

We thank the reviewer for the numerous suggestions he/she made, especially to explore the “nuances”. While we took into account most of them, we decided not to include some analyses the reviewer suggested (particularly at the within geographic area scale) because of a lack of statistical power due to a limited number of accessions (only four for Asia and Middle-East); in order to avoid any risk of over-interpretation of our results.

More specifically,

1a) It seems strange that the authors analysed all traits univariately, and then presented a multivariate PCA. Is there a justification for not continuing the multivariate analysis to compare differences in competition and geography?

The PCA was mainly carried out to provide the reader with an overview of our (rather complex) dataset. Multivariate analysis of variance requires some important assumptions to be fulfilled that we were violating in the present case (e.g. the dependent variables should be normally distributed within groups; only two of the life history traits we analyzed satisfied this criteria). We have therefore chosen to analyze each trait independently when estimating the effect of each explanatory variable. This approach also facilitates the interpretation of the effect of the density of competitors and of the geographical origin of the accessions on the various life history traits.

1b) Relevant to this, the authors cite phenotypic trade-offs as responsible for the observed patterns (e.g., L.35, L.381-382), but do not present any correlations among the traits to quantify these trade-offs. If correlations among traits differ between geography or competition (as the PCA suggests), this would be very interesting and support their conclusions very well.

We agree that in the previous version of our manuscript, this argument was only verbal based on opposite trends for some phenotypic traits (e.g., higher number of seeds produced, lower germination rate). As suggested by the reviewer, correlations between the different traits are now reported in the main text (see figure below; this figure was included in the manuscript as Fig. S5 in supplementary information). However, we chose to present only global correlations between phenotypic traits because we lacked statistical power to analyze trade-offs within geographical origins and/or levels of competitor density (there were only four accessions for Asia and Middle-East geographic areas). While PCA is mainly descriptive it contributed to the understanding of the trade-offs.

Fig S5 modified- Correlations between traits. Correlation was calculated on the complete dataset (i.e. all the accessions and all the competitive levels). The correlation value between each variable is indicated in black and only significant correlations ($P \leq 0.05$) are colored: positive correlations are in blue and negative correlations are in red.

2) The principal components analysis is very interesting, it seems as though the effects of geography are orthogonal to the effects of competition. This would suggest that competition acts upon different traits compared to expanding to new geographic areas. I think this needs to be highlighted, and tested specifically using a two-way MANOVA for example (there is a natural groupings in the data so PCA may not be the best method, and doesn't give a statistical test). On this note, Fig.1 was also difficult to interpret (especially without a legend), is it possible to use colours for region, and then different symbols for competition?

For the MANOVA, please see the commentary above.

Regarding the orthogonality between the effects of geographical origin and competition, we fitted the following mixed linear models to the coordinates of principal component 1 and 2:

$$y = C + G_o + (C \times G_o) + |B| + |A|$$

(C is the number of competitors, G_o the geographical origin, B the block effect and A the accession, the latter two being set as random effects).

The model fitted on PC1 coordinates showed a strong effect of the geographic origin but no effect of competitors, while the model run on PC2 coordinates showed the reverse pattern (see below Anova table). However, given the comments from the other reviewers about the complexity of the results we decided to not present that analysis in the paper, arguing that both the PCA plot and the relative contribution of the variables to the axes were showing that result.

Anova Table

Response variable	explanatory variables	Chisq	Df	Pr (>Chisq)
PC1	Go	85.24	2	< 0.001
	C	6.54	4	0.16
PC2	Go	0.29	2	0.87
	C	105.45	4	< 0.001

We modified the PCA figure and changed the legend accordingly in order to make it clearer (now, Fig 1c). We also added a graph representing the relative contribution of each variable to the principal components. It now clearly appears that the variables that explain most of the variance between geographic areas and between the number of competitors differ (respectively, *life time* and *flowering start* for the former and *number of fruits* and, to a lesser extent, *flowering time* and *growth rate* for the latter). The main text was also amended (L: 260).

Finally, we also added the following figure as supporting information (Fig S2). The first panel presents the relative contribution of the variables to the principal components. The second and third panels are the two first principal components but the individuals are colored and grouped according to geographic origin or number of competitors, respectively.

Figure S2 - Principal components analysis based on *C. bursa-pastoris* life history traits.

3) Noting from Fig.1, I was wondering if the authors checked for differences in variance? It seems as though European accessions have higher variance in both PC1 and PC2, which would be interesting because the invasion front could be associated with reduced phenotypic variance due to strong selection, inbreeding or selfing. This also contrasts with Fig.2, where Asian accessions showed higher variation in response to competition, suggesting that inbreeding/drift could determine expansion potential. I feel as though these could be results as well.

We thank the reviewer for that suggestion. European accessions have indeed a higher variance for PC1 (2.4) than the two others main origins (Asia, 0.8 and Middle-East, 1.1), but it also has the lowest variance for PC2 (1.4) and Asian accessions have the highest (1.6). These results tend to support the reviewer hypothesis and some of our conclusion; we are not sure, however, that they provide strong-enough evidence to build upon.

4) The use of likelihood ratio tests seems strange (e.g., L245-L249 & L278-284). Are these univariate tests removing the variable of interest? Why not

just report the results of the ANOVA and linear model summary with post-hoc pairwise comparisons?

To test for differences between geographic origins, we applied the principle of parsimony for model simplification, i.e. a variable was retained in the model only if it causes a significant increase in deviance when it is removed from the current model.

We realized a stepwise simplification using likelihood-ratio-tests following Crawley's recommendation (R book (2007): §"Statistical modelling" and "Regression"): i) We first removed non-significant interaction terms. ii) We then removed non-significant explanatory variables and iii) we finally grouped together factor levels that do not differ from one another.

5) I was a little disappointed not to see the proportional hazards curves presented (Table 2). These could help to visualise how the regions differed.

The proportional hazard curves were presented for the germination in soil in Supplementary information. Following the reviewer's recommendation we moved Table 2 to supplementary data (now, Table S5) and instead present the hazard curve as a main figure (Fig. 2).

I also had some concerns over how natural this experiment was, considering another study (Yang et al. 2018) seemed to have done the same test and find different results. Here, the authors conclude this is due a difference in methodology (L.422-429). If competition changes depending on whether seeds are planted at the same time versus three days later, then this has important and has significant biological meaning. This needs to be better justified.

The experiments were conducted under similar lab conditions, i.e. in growth chamber at 22°C and 12:12h light:dark cycles. However, some differences between the two experiments can lead to different results. First, the accessions used in Yang et al (2018) and in our experiment are different. Only few accessions are common between the two studies (and if we compare the common accessions, we find similar trends). Second and more importantly, the difference was due to the fact that the measures were not recorded at the same moment: start of senescence in both Yang et al. (2018) and Petrone et al. (2018), whereas we recorded the data after the death of the plant in the present study. We discuss these points in more details line 423

Minor comments

There is some vague language at the start of the introduction. For example, the use of 'ecological processes' (L.42) does not say much, when something like 'suitable habitat' is more descriptive and accurate. Same for 'genetic factors' (L.40). Also, on L.53 'alternatively' doesn't follow from the previous argument. I think this first paragraph should be split and written more clearly.

We have edited the first paragraph.

L.131 Could you tell the reader how many seedlings per accession on average?

Done. See line 128.

L.189 I found the equation difficult to follow. It would be better if the authors used more simpler letter notation to denote each term. For example, $C_j \times G_k$ for the competition-by-geographic origin.

Done. See line 200.

L.191 & L.214 The use of experimental block as a fixed, rather than random effect should be justified. It makes me wonder if they were avoiding random effects models.

In literature is still exist the debate about random or fixed effect concerning block effect.

Exactly, there is still a debate in the literature on whether block effects should be considered random or fixed. Treating the block effect as random has been recommended for balanced incomplete block designs because it results in smaller variances of treatment contrasts. Here, we realized a complete randomized design (each block is composed by the same treatments and the same accessions) and there is no evidence that it would be better to treat block as a random effect. Actually, Dixon (2016) still argues that it is better to model the block effect as a fixed effect.

Ourselves we do not have strong views about what is best between fixed and random effect and we conducted both types of model to see whether this led to differences (see table at the beginning of the answer to reviewers). Considering the block effect as random did not have any major effect on the results. Since referee 3 also suggested to use GLMM and to consider the block as a random effect we did so. Materiel and Methods and results sections were modified accordingly.

L.197 & L.258 What are 'sub-models'? This should be described better

Done. See line 208.

L.210 To make this equation more accessible to the readers, authors could describe it more simply as the number of fruits standardised by the mean in the absence of competition (or something similar).

Done.

L.286 'weighted' should be 'weighed'

Done.

Referee: 2

Comments to the Author(s)

Summary: The authors' manuscript, "Ecological strategy and genetic load in the shepherd's purse (*Capsella bursa-pastoris*) from the core and the limit of its natural range", presents an interesting and extensive study on the interaction of competition with genetic load and fitness proxies across the species range of an autotetraploid throughout its lifespan. I commend the authors for what must have been an immense amount of work in the greenhouse with the multi-generational, lifespan measures and data collection that went into this study. I think this is great work that offers valuable insights to the field. I have several comments/questions/suggestions for understanding and improving the manuscript.

Thanks

Major points:

1. I think a major point (if not the main major point) the authors try to make at a broader scale in this study is that it is necessary to measure different fitness traits throughout lifespan to get an accurate picture of 'true' fitness (long-term reproductive success). If I am correct in my assessment, then I am confused as to why the competitive index only takes into account the number of fruits produced. I see that this has a historical definition, but can what the authors deem to be a better measure of fitness (offspring germination success, or some combination of traits?) be used as a comparison competitive index. We could then see if this does indeed change the overall story in terms of whether simply counting fruits is sufficient or not to represent fitness, or if a novel competitive index

that accounts for multiple important fitness-related traits is similar enough that future studies would suffice to only count fruits.

We fully agree with the reviewer's comment. We have conserved the I_c (Competition index), first to be able to compare our results with previous studies, but also because this index was computed to quantify a direct effect of competition on the focal plant. However, in order to stress the necessity to consider several important life history traits to better estimate plant fitness we also computed a relative fitness index, as suggested by the reviewer. F_{index} is an integrative measure of *fertility* (*Number of fruits*, N_F) and *germination rate* of the progeny (GR_{soil}). It thus takes into account two main fitness components corresponding to different stages of the life cycle:

$$F_{index} = GR_{SOIL} \times N_F$$

We included a summary-figure in the main text (Fig 1d). It confirmed that accounting for more than the number of fruits when calculating fitness does really matter, showing that inferences based on a single trait can be misleading.

Moreover, we also have also computed the competitive Index using the Fitness index instead of the sole number of fruits (i.e. $I_{C2_{ki}} = \frac{F_{index_{ki}}}{\bar{F}_{index_{k0}}}$). We obtained the same result than that based on the number of fruits, meaning a strong effect of the geographical area ($\chi^2 = 11.12$, $df = 2$, $P < 0.01$). Asian accessions were less sensitive to the competition than European and Middle-Eastern ones (see below the summary of the model). However, for the reasons exposed above and since the results of the two analyses are similar, we decided to keep the I_c based on the number of fruits in the main text.

Summary of the model for I_{C2}

	Estimate	Std. Error	z-value	Pr (> z)
intercept	0.83	0.35	2.37	0.018 *
EUR	-1.17	0.41	-2.89	0.003 **
ME	-1.37	0.44	-3.11	0.002 **

2. I think the genetic load estimates play a major role in this paper (the title includes mention of genetic load), but I found the methods description of this aspect to be very short and entirely require going to the previous study that is cited. I think sufficient explanation should exist within this manuscript for readers to understand exactly what was done to obtain the load measures used in this paper. For example, the equation in lines 177-178 uses the terms del_{Co} and del_{Cg} which are not mentioned elsewhere in this paper and therefore confusing.

We agree and a specific section dedicated to the estimation of the genetic load was added to the Material and Methods section in §"Genetic load estimation (L. 174). We also took care to specify each abbreviation that was in the main text.

Table S1 which is cited as well seems to instead be a list of sample locations, so I am not sure what was meant to be referred to here?

Indeed, it was not well referred. We decided to add the values of the genetic load for each accession in Table S1.

I lack clarity in the main text as to whether all accessions used in this study are the same as those in the study cited for deleterious load proportions might be needed. I was initially confused in line 177 where it says load was the average proportion, which I thought meant across accessions, but this is across sub-genomes, so it should be correct that load is on a per-accession basis? And deleterious sites are identified using SIFT4G on nonsynonymous mutations in each accession? I would mention the use of SIFT on these sites in the main text here.

As mentioned above it is now clearly stated in the Material and Methods that: i) Genetic load was measured by Kryvokhyzha et al. (2019) for each accession and each of the two subgenomes of *Capsella bursa-pastoris* independently (thus two values per accession) and ii) in that study, for each accessions, we considered the average of these two values as the global genetic load.

3. There are a lot of results in this study, and I find it difficult to digest all of them. Along these lines I have several comments:

We have added a summary table with the traits, their abbreviations, and the main trends for each trait. See Table 2.

3a. In lines 236-240, how is GR_MP different from GR_AGAR and GR_SOIL? Line 146 is the first mention of GR_MP with no definition, and then it next appears here in line 236, but I do not see what it is nor what MP stands for?

GR_MP (MP stands for "mother plant") corresponds to the germination rate, in agar, of all the accessions that were used in our study (i.e. before they enter the experiment). GR_AGAR and GR_SOIL correspond to germination of seeds of the progenies grown on agar and in the soil, respectively. Some accessions were not used in our competition experiment because of a too low germination rate, GR_MP (see Table S2). We added a definition of GR_MP (L. 136)

3b. Is the negative correlation of GR_MP with deleterious load only in Middle-Eastern accessions, or is it across all accession? It could perhaps be nice to see these split into region (Table S3 is all accessions not divided into region), but see my next comment.

It is across all accessions and we modified the text to make it clear (L. 264). For the very same reasons for which we did not look at phenotypic trade-off within geographic area (see answer to referee one point 1b), we did not study correlations within geographic area because of a too low number of accessions in some cases (four accessions for Middle-East and Asia).

3b. I am not sure how much I learn from figure 1, in case there are figure limits, but I did think it might be useful to have a multi-panel figure showing each trait measured (germination rate, growth rate, ...) as compared to the genetic load per individual accession. Accessions could then be colored by region and significant correlations could be highlighted by, e.g., thicker regression lines on the plot. Number of competitors could be indicated by point size, but then I begin to add more confusing levels... I am imagining something along the lines of Figure 3 in Bosshard et al 2017, Genetics which would let the readers immediately see an overall view of what traits correlate and in which directions. I imagine this might allow for a significant reduction in the text for the results, or at the very least a clearer overall picture of the direction of correlations for ecological traits with load across the accessions (and thus across geographic space as well). From Table S3 I think this could be very neat because it looks like the majority of significant correlations are negative, except for flowering time and growth, which could make sense if very loaded individuals "live fast and die young" in order to have any reproductive success at all.

Along with referee 1, we think that Figure 1 is really informative as it highlights the fact that both geographic origin and competition intensity strongly determine phenotypic variance by acting on different phenotypic traits. We draw a completely new figure to highlight the key features of the PCA (Fig. 1 and answer to point 2 of referee 1). However, we agree that a figure showing the relationship between trait values and the genetic load is missing. As suggested, we added a new figure in the supporting information (Fig S6). For each trait for which a significant effect of geographic origin was found, dot' sizes correspond to competition intensity and colors to

genetic cluster. Solid lines are for significant Spearman's correlation. Finally, we are grateful for the expression "live fast and die young" that summarizes so well our conclusions; we included it in the main text.

4. I do not see it as a fatal flaw, but I found the flow of the discussion difficult to follow when trying to keep all of the results in mind. I am not sure I have a good suggestion for improvement, so would just ask the authors to at least take a moment to ponder any potential improvement. For example line 431 begins with discussing a shift in phenology, but it is difficult to follow from the previous paragraph as that ends with several sentences referring to another study. Even just mentioning "A shift in phenology to earlier flowering at the range edge can be due to..." would have helped, as I had to go back and reread the previous paragraph in its entirety. If I continue in the paragraph starting at 431, it is a similar structure to others where the result is discussed, a future direction is mentioned, a caveat is mentioned, and a previous study is mentioned. Perhaps it is personal preference, but I think I am less prone to get lost if all the caveats across all results appear in a paragraph along with the other relevant previous studies that may make them not caveats / not relevant for the study at hand. Then when I am reading the parts of the discussion interpreting the results, it is simply interpretation and implications for the future.

The discussion was modified according to the reviewer recommendations. We have shortened the discussion and reorganized each part so that the caveats are now clearly identified and placed at the end of each part. We hope that modifications made in the Result section and the new table summarizing the results also help to highlight the main findings.

5. Following up on my first major comment, I think the last paragraph of the discussion is again a major take-home from the paper. The discussion might be better served by putting this first. I would clarify the writing in this paragraph to be clear that Asian accessions have the most genetic load, yet the highest fitness measured by fruit count - a contradictory result, until you look at your other fitness measures (seed quality or others?) that show the lowest inferred fitness (thus also go back to my major comment 2 in relation to this).

We agree with the referee. We have now restructured the discussion and this part was moved to the beginning. We tried to clarify the writing to avoid confusion and to highlight the take-home messages of our study.

6. Do any caveats about genetic load in a tetraploid need to be mentioned? Maybe this was discussed in the previous paper. But maybe masking of deleterious recessive variants due to the larger genome size could be a relevant phenomena that I do not think is able to be addressed in this study. Unless the authors also want to re-do analyses with the proportions in each sub-genome independently, but I think a simple mention of this caveat may be sufficient.

As the reviewer pointed out we will not be able in the present study to address the fact that we are working with a tetraploid and that the effect of some deleterious alleles may be masked so we did not discuss this issue. We noted this issue in the discussion though.

Minor points:

1. I would add a legend of the point colors directly on the plot in Figure 2 rather than necessitating reading into the caption.

We used the same color scheme for all the figures. This is why we did not feel it would be useful to add a legend directly on the last figure of the paper.

2. In Figure 2B and 2C, I take μ in the y-axis to indicate mean, but it is not immediately stated in the caption. On the x-axis of 2B, I think it should not be a μ , but is rather a proportion? It would be one less step for readers' thinking if 'mean' and 'mean proportion' were written out (I think there is sufficient space).

Done. See Figure 3

Referee: 3

Comments to the Author(s)

In this manuscript, Orsucci et al. use accessions sampled from three geographic regions with differing colonization history to assess phenotypic evolution during the recent invasion of *Capsella bursa-pastoris* into Asia. Their data suggest that populations from the more recently colonized part of the range have evolved distinct ecological strategies through changes in several fitness-related traits. Furthermore, they show that the earlier flowering phenology of recently-colonised Asian populations makes them less susceptible to interspecific competition than accessions from other regions. In my opinion, the most novel aspect of this work is the focus on tradeoffs among traits representing the full life cycle and their ability to connect this to competitive ability. The described associations between the phenotypic data and estimates of genetic load are interesting, but, as they note, do not disentangle the confounding effects of genetic load and geographic area (lines 372-377). Nevertheless, this work presents an interesting attempt to combine ecological, phenotypic and genetic data to better understand the processes underlying recent range expansions.

Thanks.

Generally, the paper is well-written and the main findings clearly highlighted, though I have a number of question and comments about certain aspects of the manuscript. My main comments are as follows:

1) I found it difficult to identify which results support the statement that "accessions from the front performed worse for most life history traits than those from the core" (e.g. lines 30-31, 101-102, 347-348). For example, progeny germination rate was clearer poor for accessions from recently colonized populations, but fruit production is clearly higher for thee same accessions. It is also not clear to me that variation in other traits (such as lifespan) can be classified as 'better' or 'worse'. To better highlight this key finding, I would therefore suggest the following changes: (a) more clearly highlight in the results section those traits most relevant for assessing fitness e.g. fruit production and progeny viability; (b) Replace figure 1 (PCA) with a modified version of Figure S1. Instead of the photos, I would include two plots highlighting variation in the most relevant life history traits (fruit production and progeny viability). To illustrate effects of competition, you could select a single competition level to compare to your control treatment in these plots. (c) Where possible, provide relevant means and standard errors in a table form (including sample sizes), rather than the text. I understand that space is limited, but you could then delete this information from the results, making this section easier to read.

In our revision we have attempted to clarify the results part, which, we agree, was a bit hard to follow. Regarding the PCA (Fig. 1b) please see commentary to referee 1 point 2 and referee 2 point 3b. Briefly, we feel that the PCA describes accurately the phenotypic variance and discriminates well the two most important explanatory variables of our dataset: the competitor density and the geographical area of origin. We have now rewritten the caption, edited the text and modified the figure to stress the most salient features of the PCA. We also followed your suggestion and modified figure S1 and Fig. 1. To highlight directly one of our take-home messages, we used

3 plots representing fertility, viability and the combination of those two traits in an integrative measure called relative fitness index (please see response to point 1 of referee 2). We also added a new table with summary of the results for the various traits (see response to point 3 of referee 2). Finally, we agree that too many statistics were provided in the main text. We now instead refer to tables and figures, which were already present in the main text and supplements.

2) Given the aim of the paper to link genomic and phenotypic characteristics of the accessions, I do not understand why so many levels of competition were used. Would it not have been better to focus on one level of competition (e.g. 4 competitors vs no competition) and use more replicates per accession (or accessions per geographic area) to better estimate fitness-related traits? Could you please justify this experimental design in the introduction and methods?

Linking genomic and phenotypic characteristics was not the main aim of the paper and, the genomic side is only considered through the genetic load. The primary aim of the paper was to relate geographical origin, taken here as a proxy for distance to the core of the distribution, and fitness. The genetic load was indeed seen here as one aspect of the distance to the core of the distribution. In retrospect we could indeed have considered less competition levels and more accessions. As a matter of fact, we did plan to have a larger number of accessions but some germinated poorly. The choice to have more than two levels of competition was made to have a more quantitative measure of competition and because we did not a priori know which level of competition would be necessary to see a response of the focal plant.

3) I have several questions about the statistical analyses performed. (a) Given accession was nested within geographic area, how in your analyses did you account for pseudoreplication? I think generalised linear mixed effects models (GLMMs) would be more appropriate in this case than the GLM you use (lines 185-195). GLMMs would allow you to include the random effects of both 'block' and 'accession nested within geographic area'.

It is always difficult to know whether it is better to consider the block effect to be random rather than fixed. In our case, considering a random block effect led to similar results than when we considered it as fixed (see Table 2 in our reply to the editor). We now consider it as a random effect across the paper.

We agree that block effect is not interesting when considered fixed and considering the variance due to the block effect seems a better approach. Consequently, we have used both glmm and lmm in the new version of the manuscript.

In our understanding we are not in a case of pseudoreplication. We have four perfect replicates containing the same accessions that are grown with different competitor densities. Although the accessions of the same genetic cluster are more similar than accessions belonging to different clusters, we do not think that this is a case of pseudoreplication. However, we agree that the accessions could influence our results and, consequently, we followed the recommendation of the referee and we added the accessions in random effect.

(b) Lines 221-222: I am not familiar with this approach of merging factor levels to simplify models. If I understand correctly, this is quite different to the standard methods of comparing factor levels using posthoc comparisons or defining contrasts in the context of more complex GLMs. Can you justify why you adopted the approach you did?

This is a posthoc comparison that we have conducted manually following recommendation in The R book, (Crawley, chapter "regression"). And it is similar to the standard stepwise regression method.

Minor comments and suggested edits:

Line 42: expand this to "ecological and evolutionary processes."

We have clarified this sentence as follows: "colonization success also depends on other biotic (e.g. species composition) and abiotic factors (e.g. temperature, moisture)" L. 43.

Lines 53-57: The topic of these sentences seems slightly disconnected from the rest of the paragraph. I think you should end the first paragraph at line 53 with a statement about the knowledge gap your study aims to test. Then you can start a new paragraph providing some more background on competition and range expansion.

Done

Lines 58-63: Again, this paragraph seems out-of-place and interrupts the flow of the introduction. Could you instead mention the benefits of allopolyploidisation later when you introduce your study system?

We now mention the advantages of Cbp (polyploid genome and selfing system) to study competitive abilities at the end of the introduction (L.77).

Line 88: Could you add an additional sentence describing the work conducted in the reference [24] - such information would be useful to better understand the novelty of the findings in the present manuscript.

Done. Line 81.

Line 152-3: "The growth of rosette size..." should be replaced by "Rosette growth rates..."

Done.

Line 170-172: In the results you refer to data on the start of flowering for progeny, but you do not mention the collection of this data in the methods. Can you add these details? Furthermore, given the progeny plants started flowering, why did you not additionally collect data on progeny fruit set?

Done. Line 170.

We measured flowering start only on treatments with 0, 2 and 8 competitors. This was done in order to test (i) whether we still observe a shift in phenology among geographical areas, and (ii) if the stress due to competition could influence the next generation (i.e. maternal effect). We observed no differences and the plants showed phenotypes similar to the phenotypes of their mother (no clear differences in number of floral hamp etc.), so we did not consider it necessary to recount the number of fruits (which is quite fastidious).

Line 225-231: I would remove this summary, as the PCA is not easy to interpret (see comment above) and the approach is not described in the methods. You could keep the sentence beginning "For clarity...", as I agree that this is a useful way of presenting the results.

We hope that the PCA is now clearer and more informative in the revised version. We decided to keep the summary which provides an overview of the most important results. As suggested by the referee, we have added a description in the Method section (L. 186) to describe the PCA method.

Can you present the results for associations with genetic load in one paragraph at the end of the results section? You could use a subtitle like "Associations between genetic load and fitness-related traits", and this would make it easier to compare the different traits. If you have space, you could also consider moving part of Table S3 (with Spearman's rank results) into the main text. Then, you could also remove the associated statistics from this written section.

Unfortunately, we have not enough space to move Table S3 to the main text and we also added a figure in supplement (Fig S5) showing the correlation of the main traits with the genetic load.

I think it is clearer to avoid describing methods in the results section. Could you remove or move the following sentences to the methods section (e.g. lines 240-242; 278-279; 286-288)?

We remove the sentences: 278-279; 286-288. But we kept the sentence lines 240-242 which corresponds to a result.

Line 300: "Complex phenology" is a confusing term - I would instead label this as "Progeny germination rates and reproductive phenology"

Done

Figure 2: Can you overlay standard deviations (or boxplots) onto the plots? It is difficult to assess the differences among geographic regions with just means.

We think that is not necessary and could overload and complicate the figure. Moreover, each dot represents an individual and by consequence highlights well the data dispersion around the means.

Table 2 is not that informative - can it be moved to the supplementary and replaced with Figure S4?

Done

Line 379: "...the lowest vegetative growth and reproductive output..." - But the Asian accessions had the highest reproductive output as assessed by fruit production. Be careful with your phrasing here.

Done

Lines 395-408: I would remove this section, as it does not seem relevant to the main topic for discussion - it is simply a (nice) overview of the transition from outcrossing to selfing.

Done. We agree and we have now removed this part.

Line 412-413: Can you provide references to support this statement?

Done.

Line 450-455: Are there any references to support this line of argument? I am not sure I understand how the "shift in phenology... may simply be a direct consequence of high genetic load". If there are no references, I would rephrase this line of argument.

Some studies (Shaw et al. 1998; Ellmer & Andersson, 2004) have shown that inbred plants flower later. Thus the genetic load can influence different life history traits and can thus change the flowering periods in plants.

Line 466-478: I agree that studying the whole life cycle can be particularly informative and is one of the key strengths of this study.

Thanks

Supplementary files:

Figure S3: Why are not all combinations of competitor treatments and traits presented here? For example, there is no plot providing fruit number for each geographic area. It might be useful. Can you provide these plots here or explain why they are not included in the figure legend?

We had only represented the traits for which a significant effect of the genetical cluster or the number of competitors was detected. In the new version of the manuscript, following the reviewer's recommendation, we constructed two multi-panel figures corresponding to the set of traits measured during the competitive experiment as a function of geographical origins (Fig. S3) and number of competitors (Fig. S4).

Appendix B

Associate Editor

Board Member

Comments to Author:

Many thanks for your extensive revisions of the MS. All reviewers and myself agree re: the value and importance of your study. Please ensure that the outstanding issues are fully addressed (and explained) in your revised submission. In particular, one reviewer would like you to make clearer the hypotheses you are testing in your study, in order to frame your findings in a more general context. It would also - in my view - be useful to make your title more informative - in terms of what the key finding is of your study, and its broader relevance. Currently your title seems rather descriptive, and could perhaps be made more informative.

The reviewers also note that at the moment you have not uploaded or made accessible the data used for this study. This needs to be addressed before publication, please.

Dear editor,

You will find below, in blue, the responses to the reviewers' comments. We have edited the text to make clearer the hypotheses tested and we also modified the title following your suggestion:

'Shift in ecological strategy helps marginal populations of shepherd's purse (*Capsella bursa-pastoris*) to overcome a high genetic load'

The data have been uploaded on Dryad:

[doi:10.5061/dryad.vt4b8gtp7](https://doi.org/10.5061/dryad.vt4b8gtp7)

<https://datadryad.org/stash/share/RZrUe1IXJQzsRYjanyhCXwTSBVosr87lpu8eMRDFV3Y>.

Reviewer(s)' Comments to Author:

Referee: 1

Comments to the Author(s).

The authors present an interesting common garden experiment to test whether populations from the invasion front differ in phenology and competitive ability to those from the natural distribution. They find very interesting evidence that although fitness is overall lower at the front, these plants possessed better competitive ability.

Overall, I think the authors have done well to address the comments made in the previous review and I appreciate the level of detail in their response, especially in their adjustments to the discussion. I have no major criticisms, just some minor comments below.

1) The first two paragraphs do not frame the topic very clearly. For example, L.52-55, the use of 'peculiar demographic dynamics' is not required, and the authors mention "allele surfing" and "expansion load" without really defining what these mean. I think the introduction could read better if the first two paragraphs framed the topic a bit more clearly. The second paragraph only highlights what has been done before and does not make any statement about what is important to understand, for example, it could benefit from outlining the importance of understanding how genetic load and competition affects fitness at the expanding edge of the range.

We have edited the first two paragraphs. It is true that there was a remaining ambiguity on what was tested which is the relationship between the evolution of the phenotype and the load during species expansion. We hope we succeeded in framing the topic more clearly and improving the link between the two paragraphs.

2) Further to point 1, I think when describing the experiment, the authors could better highlight the

hypothesis they are testing and what their predictions are. At the moment, the introduction is a little descriptive of the study system rather than focusing on the theory they are testing.

We have tried to be more explicit on what is tested.

3) L.200 the use of vertical lines to denote random effects are not (in my opinion) a good idea as these mathematically signify absolute values. I suggest using upper case for fixed effects and lower case for random effects if the authors wish to clearly differentiate the two.

Done

Referee: 3

Comments to the Author(s).

In this manuscript, Orsucci et al. use accessions sampled from three geographic regions with differing colonization history to assess phenotypic evolution during the recent invasion of *Capsella bursa-pastoris* into Asia. Their data suggest that populations from the more recently colonized part of the range in Asia have evolved a distinct ecological strategy. Specifically, the Asian populations show reduced overall fitness, consistent with the costs of an elevated genetic load, but a shift towards earlier flowering that makes them less susceptible to interspecific competition than accessions from other regions. The authors have made significant efforts to improve the manuscript based on previous reviewer comments, and I think the introduction and results are now much easier to digest. The new analyses with the mixed models are consistent with their original analyses, and the figures and tables in the main text now better highlight the main findings of the manuscript. I only have a number of minor comments that I think would help improve the clarity of the manuscript and particularly parts of the discussion.

Minor changes

Introduction, Methods and Results:

Line 33: I find the use of the phrase “performed worse for most life history traits” unclear, as it is not apparent what “performed worse” means. Given you produce a combined fitness index, I would simply state “showed reduced fitness” here and elsewhere in the manuscript (e.g. line 357).

Done. We have modified the abstract and the first sentence of the discussion to be more general. However, we would prefer keeping the term “performance” as we only approximated the fitness through two of its main components.

Line 101: change to “the highest genetic load” or “a high genetic load”

Done.

Line 102: perhaps replace “very early flowering start” with “early initiation of flowering”

Done.

Line 259: you could be more specific with your subtitle here: “Reduced germination rates, but earlier flowering”.

We modified the title as : « Reduced overall performance but earlier flowering of accessions from the colonization front ”.

We decided to keep the general term of performance because in this part, we describe all the life history and phenological traits, not only the germination rate.

Line 267-267: I would rephrase this sentence: “European or Asian accessions grew at similar rates, but Middle-East accessions showed a much faster growth rate”

Done.

Line 273: should be “competitors”

Done.

Line 278: replace “lowest lifetime” with “shortest lifetime”

Done.

Line 298: replace “as for mother plants” with “Similar to mother plants”

Done.

Line 332: change “when the competition increased” to “with increasing levels of competition”

Done.

Line 344: I would adjust this subheading to: “Asian accessions show reduced overall fitness”

Done.

Discussion, tables and figures:

Line 367: “...considering measures of fitness that are integrative and as close as possible to fitness itself ...” I find this a little unclear; perhaps you could simplify this to: “measuring multiple components of plant fitness...”

Done.

Line 371: This subtitle makes a strong statement given that you only assess correlations among traits and genetic load, and that the number of fruits produced did not significantly differ among geographic accessions. I would rephrase this to something like: “Recently-colonised populations show reduced fitness consistent with their higher genetic load”.

Done. We replaced by: “Front populations showed a reduction in fitness, consistent with their high genetic load.”

Line 375 should read “may have been partly mitigated by...”

Done.

Line 381-382: I would suggest the following rephrasing: “... showed that reduced seed number was offset by increased seed mass and survival”

Done.

Line 393: this subtitle is not very descriptive – could it become a concluding statement? For example, “Phenological shifts in recently-colonised populations reduce effect of competition”

Done.

Line 394-395: Rephrase this sentence: “The reduced mean fitness we observed for Asian accessions could also be offset by their enhanced competitive abilities

Done.

Line 397-402: I don’t really see the relevance of discussing priority effects here. Unless I misunderstand the term, your data does not show that these early flowering accessions shape subsequent community assembly. I would therefore replace this section with a simple (re)statement of your finding that accessions from the colonisation front flowered earlier than those from the range core.

We prefer keeping that point in discussion. We think that it is important to point out for the reader that it is common in invasion fronts that individuals experience a shift in phenology that facilitates their settlement in the new environment. Even if we did not specifically test for a priority effect, our results are consistent with that theory.

Lines 393-445: To break up this currently long section and make it easier to follow arguments, I would start new paragraphs at the following locations: line 405 (“A shift in phenology...” and 416 (“There are some important caveats”)? And 429 (“We have so far discussed ecological explanations”).

Done. Following the reviewer’s recommendation, we split the section at line 405, but not at (L. 429) to keep all the caveats together.

Line 405-407: The distinction between “high phenotypic plasticity” and “an adaptive response to different environmental conditions” is not immediately clear to me – could you expand on this a bit further?

The early flowering can be explained by (1) a high plasticity of the trait of individuals in colonization front (the more plastic you are the easier is the installation), (2) an adaptive response through directional selection: individuals that bloom early are selected for since it is advantageous to flower early and it eventually shifts the phenology.

Line 411-412: I am not sure I understand this sentence – perhaps remove?

Done. We removed this sentence as suggested by the reviewer.

Line 427: change to “very low germination rates”

Done.

Table 1: What does the final column refer to? I presume it is the error term, but it is best to explain that here in the table legend.

Done.

Table 2: I think this is a useful table to summarise the key results for the many traits measured, but I would rephrase the table title: “Summary of differences in key life history traits across the three geographical groups”. I would also change some of the column headings: replace “main trends” with “significant trends” and “additional comment” with “main conclusion”. Note that in the final column

“than ME and EU” is repeated twice in one box.

Done. The name of the column ‘main trends’ was kept as all the trends are presented, not only the significant ones.

The figure legends appear to be missing from the main text.

Done. We have added the figure captions in the last version

Referee: 2

Comments to the Author(s).

I apologize for my delay in returning this review; I was overly optimistic about my productivity levels during the coronavirus lockdown and unwisely accepted several manuscript reviews, putting this one later in the list.

After reading the revised version of the authors' manuscript as well as the comments in response to my previous reviews, I am happy to say I am satisfied with the responses and edits. I really think this is a great study, and am glad that it reads more clearly and cleanly now. I have no further comments or edits to request.